



# TRACING SUSPENDED SEDIMENT FLUXES USING A GLIDER: OBSERVATIONS IN A TIDAL SHELF ENVIRONMENT

Sabrina Homrani[1,2], Orens Pasqueron de Fommervault[3,4], Mathieu Gentil[5], Frédéric Jourdin[1,6], Xavier Durrieu de Madron[7], and François Bourrin[7]

[1]SEDIM laboratory, Shom, Brest 29200, France
[2]ENSTA Bretagne, STIC/MAD, Brest, 29200 France
[3]ALSEAMAR company, ALCEN group, Rousset 13790, France
[4]OceanOPS, World Meteorological Organization/Intergovernmental Oceanographic Commission of UNESCO, Monaco
[5]Aix-Marseille Univ., Université de Toulon, CNRS, IRD, MIO UM 110, Marseille, France
[6]Geo-Ocean laboratory, UBO University, Plouzané 29280, France
[7]CEFREM laboratory, UPVD University, Perpignan 66000, France

**Correspondence:** Sabrina Homrani (sabrina.homrani@gmail.com)

**Abstract.** Underwater gliders equipped with current profilers and optical turbidity sensors offer a low-energy solution for high-resolution measurements of currents, suspended particle properties, and sediment transport in coastal waters. Because the spatial structure of hydrosedimentary processes often changes on short time scales (hours to weeks), especially in coastal areas, validating the distribution of glider observations is required to assess our capacity to represent hydrosedimentary pro-

5 cesses. Here we propose to validate in a shelf tide-dominated environment, both i) glider-based currents, and ii) glider-based acoustic backscatters and optical turbidities in full resolution delayed mode, using in situ colocated and synchronous ancillary observations. The deployed glider system correctly measures the periodic pattern of the tidal current, with a satisfying RMSD of $O(3 \text{ cm s}^{-1})$. Glider optical turbidities highly correlate with the ancillary observations ($R^2$ up to 0.83). They also correlate well with their glider acoustic counterpart for most of the campaign period ($R^2 = 0.76$), allowing an estimation of suspended

particulate matter concentrations from the acoustic. In this study, we showed the presence of bottom nepheloid layers of several mg·l[-1] on the shelf probably due to advection of coastal turbid waters as evidenced by estimated glider sediment fluxes. These results highlight the potential of gliders for quantifying sediment fluxes and advancing our understanding of coastal hydrosedimentary processes.

## 1 Introduction

To advance ocean monitoring efforts, Essential Ocean Variables (EOVs) have been identified as critical metrics for understanding energy and matter transport across the land-to-sea continuum by the Global Ocean Observing System (https://www.goosocean.org). In coastal regions, where rapid temporal and spatial variability governs material exchanges, EOVs like currents and suspended particulate matter (SPM) are pivotal for sediment transport, carbon cycling, and nutrient dynamics (Durrieu de Madron et al., 2008). Accurate observation of these variables is essential for predicting sediment fluxes and assess-





ing environmental changes under increasing anthropogenic and climatic pressures (Ouillon, 2018). However, capturing the full dynamics of the water column in coastal zones, such as continental shelves, remains challenging due to the highly dynamic nature of these environments and the limitations of traditional tools like moorings, satellites, and research vessels in providing adequate spatio-temporal coverage.

Over the past decade underwater gliders equipped with advanced sensors—including acoustic Doppler current profilers (AD-CPs) and bio-optical instruments—have emerged as promising tools to address these challenges (Glenn et al., 2008; Bourrin et al., 2015; Miles et al., 2015, 2021; Many et al., 2016; Gentil et al., 2020, 2022). These autonomous, torpedo-shaped platforms are capable of diving up and down the water column by adjusting their buoyancy (Davis et al., 2002; Rudnick, 2016), enabling them to collect high-resolution temporal and spatial data over long deployment periods. By complementing traditional observation systems, gliders offer a unique opportunity to resolve the full water column dynamics in coastal environments.

Despite these advancements, validating glider-derived currents, remains a challenge. The typical accuracy of glider-ADCP measurements is about 0.05–0.1 m s$^{-1}$ (Ordonez et al., 2012; Todd et al., 2017; Heiderich and Todd, 2020; Jakoboski et al., 2020; Gentil et al., 2020), assessed using methods such as geostrophy balance, bottom tracking, or depth-averaged currents (DACs). While these approaches are useful, they often present biases in coastal zones where numerous processes—such as tides, internal waves, and wind-driven currents—interact and create significant spatio-temporal variability. Using in situ col-located and simultaneous Eulerian measurements has been shown to provide the most reliable validation of glider-derived currents, as reported by Thurnherr (2010) and Ellis et al. (2015). This type of validation remains rare due to the difficulty in maintaining instrumentation over the shelf, given the risks associated with intensive trawling activities (Ferré et al., 2008). Furthermore, the above-mentioned studies have, for the most part, not been carried out exhaustively when compared to the periodicity of the hydrodynamic forcing that drives the coastal currents being measured.

Beyond currents, SPM plays an equally critical role in shaping sediment transport on continental shelves. Optical and acoustic signals are often used as a proxy of suspended particulate matter which can vary significantly with respect to particle concentration and particle properties such as size, nature, and shape (Lynch et al., 1994). Optical turbidity for a given concentration of suspended particles increases with decreasing particle size, due to both increased abundance and to light scattering from smaller particles (Kitchener et al., 2017). ADCPs integrated into gliders operate at between 0.614 and 2 MHz, measuring the acoustic intensity of the received reflections from particles. At these frequencies, the peak sensitivity of the sensor is comprised between 250 and 775 $\mu$m (Lohrmann, 2001), which allows to measure large particles or aggregates in suspension in the water column. While glider-based measurements have demonstrated potential for distinguishing small and large sediment particles (Miles et al., 2015; Gentil et al., 2020), efforts to quantify SPM concentrations and fluxes remain limited, often hampered by uncertainties in sensor calibration and environmental variability. This limitation affects our ability to model sediment transport accurately, particularly during extreme events like storms or floods.

Given the glider observation distribution in space and time, we are left with the following questions: i) Are the full-resolution optical and acoustic observations of gliders sufficiently accurate and reliable to allow to quantify hydro-sedimentary processes in the coastal zone? ii) What are the processes involved in the spatio-temporal changes of particles properties in this study and what are the implications in terms of SPM fluxes? By addressing these questions, gliders offer the potential to not only enhance





our understanding of sediment dynamics but also provide critical data for managing and predicting coastal processes. This study investigates the hydro-sedimentary dynamics of a tidal continental shelf using a SeaExplorer underwater glider equipped with ADCP and bio-optical sensors. Combining glider observations with co-located moored ADCPs and rosette sampling, we aim to validate glider-derived currents and SPM estimates over multiple tidal cycles. This approach seeks to improve the accuracy of sediment flux quantification in dynamic coastal environments.

The article is organized as follows: Section 2.1 presents the field campaign while Section 3 enumerates the instrumentation used. Section 4 describes the processing methods applied to recorded data and Section 5 displays and discusses the results obtained for currents, Suspended Particulate Matter Concentrations (SPMC) and SPM fluxes, with a focus on the accuracy of glider observations and the hydro-sedimentary processes driving spatio-temporal variability in particle properties. Finally, a conclusion draws a perspective for this work.

## 2 Field campaign

A survey entitled MELANGE happened between 0730 UTC 14 February 2021 and 1300 UTC 18 February 2021 on the French Armorican shelf. Subsection 2.1 presents the studied area, while Subsection 2.2 describes the environmental conditions encountered.

### 2.1 Study Area

In order to validate the glider observations of the ocean currents in a meso-tidal area, the French Armorican shelf has been chosen because it is a well-known study area where the semi-diurnal tide is the main hydrodynamic forcing (Vincent and Le Provost, 1988). In order to observe suspended particle fluxes as well, the survey has been settled more precisely inside a broader belt-shaped area comprised of muds and silts known as the 'Grande Vasière' (Dubrulle et al., 2007). The geographic centre of the survey is a mooring called GV1, which is also part of a long-term (5 years) network of benthic measurements around French Brittany, called ROEC/Benth (Marchès et al., 2019). All MELANGE measurements were performed in a square area centred on GV1 and having a side length of about 10 NM (Fig. 1a). This square is located at around 25 NM from the coast, at mean ocean depths of 115 m ±4 m. The extent and location of the area were chosen on the mid-continental shelf in order to encompass along-shelf and cross-shelf gradients of ocean turbidity.

### 2.2 Sea conditions

The main hydrodynamic forcing in the validation area is the semi-diurnal tide, dominated by the M2 and S2 components (Vincent and Le Provost, 1988). During the validation campaign, the tidal range decreased from 3.9 m to 2.5 m at the GV1 mooring (Figure 2b). The south-to-north component $v$ reached its extrema in the 5th hour before slack water (Fig. 2c). The west-to-east component $u$, 2 hours approximately phase shifted, reached its extrema in the 2nd hour before slack water, with lower maxima than $v$ (Fig.2c). Consequently, the overall tidal current was maximal in the SSW-NNE directions, minimal in the orthogonal directions, and had magnitudes between 0.1 and 0.3 m s$^{-1}$.



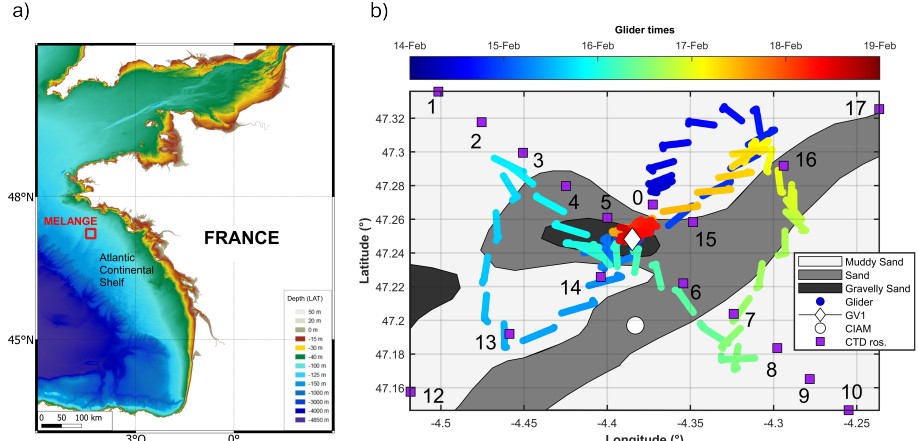

**Figure 1. (a)** The MELANGE campaign area, off the French Atlantic coast. Bathymetry from SHOM (2015); **(b)** SeaExplorer glider path with locations of CIAM (white disk) and GV1 (white diamond) moorings, and CTD-Rosette casts (purple squares), superimposed on the surface sediment composition of the ocean floor according to Garlan et al. (2018).

The wave characteristics for the validation period, calculated with the WAVEWATCH-III wave generation and propagation model (Tolman et al., 2009) and interpolated at the GV1 mooring, show mean significant heights and wavelengths of 4 m and 148 m, respectively, mostly coming from the west (Fig. 2d), and maximum values of 5 m and 245 m, respectively. According to the meteorological equipment installed on RV (Research Vessel) Thalassa, the wind blew from the SSW, with a mean intensity of 12 m s$^{-1}$ and peaks up to 23 m s$^{-1}$ (Fig. 2e). The spatial extent of the validation area is small enough (submesoscale) to consider that the physical parameters given in this paragraph (mesoscale) are homogeneous, and are thus valid at the whole study area spatial scale. Furthermore, the measurement period allowed us to sample 8 tidal cycles, i.e. 8 times the period of the dominant forcing, with variable amplitudes, enabling us to assess the performance of the glider over a representative period of the tidal forcing.

## 3   Materials

In addition to the glider (Subsection 3.2), the survey involved two bottom moorings (Subsection 3.3), and a CTD-Rosette (Subsection 3.4) onboard RV Thalassa. The first subsection 3.1 introduces the measurement strategy we adopted.

### 3.1   Measurement strategy

During the first period of the campaign, lasting from 0730 UTC 14 February to 1320 UTC 17 February 2021, the glider navigated following a BUtterfly pattern (hereafter abbreviated BU) centred on GV1 (Figure 1b). This pattern is relevant for comparing fixed measurements and glider measurements at submesoscale (Bosse and Fer, 2019; Rollo et al., 2022). During BU the glider acquired more than 170 vertical profiles at distances less than 6 NM from GV1. For the remaining period, the



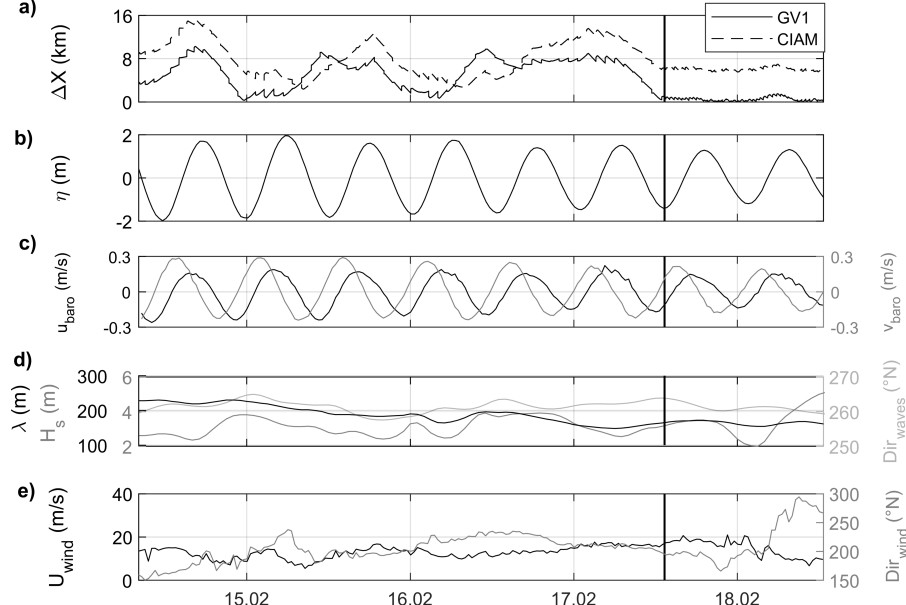

**Figure 2.** Environmental settings: **(a)** Glider-moorings geographic distance, **(b)** tidal elevation [m] from the GV1 ADCP pressure sensor, **(c)** tidal current components $u$ and $v$ from the GV1 ADCP, **(d)** significant wave height, wavelength and direction estimated from the WWIII model from Ifremer (2022) at GV1 location, **(e)** wind speed and wind direction from RV Thalassa weather station. The black line delineates the Butterfly and Virtual Mooring survey periods (cf. Section 3.1).

glider performed a Virtual Mooring (hereafter called VM) with almost 45 profiles in much closer vicinity of GV1 (at 500 m $\pm 150$ m from it), in order to acquire the most consistent data for comparison with this mooring. During the whole campaign, each glider profile lasted about 20 minutes allowing a sampling of each semidiurnal cycle with about 36 profiles. The glider vertical velocity is 0.2 m s$^{-1}$ on average.

In order to evaluate the spatial consistency of the currents, a second ADCP was moored at a distance of about 3 NM south of GV1. This mooring, whose location is also displayed in Figure 1b, is called CIAM, standing for "Châssis d'Instrumentation Autonome de Mesures". The coordinates of the two moorings are given in Table 1.

Finally, a CTD-Rosette sampler was deployed, performing an along-shelf section with 11 stations on February 14 and a cross-shelf section with 6 stations on February 18 (Figure 1b). These CTD-Rosette casts, with in situ calibrated acquisitions based on water samples, are considered as the reference measurements for temperature, salinity, bio-optics with SPM measurements especially.

The glider, CIAM and the CTD-Rosette were deployed from RV Thalassa. GV1 was previously moored in January 2021, in the framework of ROEC/Benth.




**Table 1.** Locations of the two moorings (ITRF2014 geodetic system) with their Lowest Astronomical Tide (LAT) chart datum depths.

| Name | Depth | Latitude | Longitude |
|------|-------|----------|-----------|
| GV1 | 113 m | 47°14.483'N | 4°22.611'W |
| CIAM | 117 m | 47°11.821'N | 4°22.986'W |

**Table 2.** Configurations of the downward-looking glider mounted AD2CP and the two upward-looking CIAM and GV1 moored ADCPs. The velocity uncertainty (last Table line) is the standard deviation error for both components ($u$ and $v$) of the velocity profiles. The standard deviation values has been estimated in Section 4.5 for the glider and in Section 4.6 for the moorings.

| Parameter | Unit | GLI | CIAM | GV1 |
|-----------|------|-----|------|-----|
| Bin size | m | 2 | 2 | 2 |
| Range | m | 30 | 92 | 92 |
| Blanking | m | 0.2 | 4.2 | 4.2 |
| Secondes/ping | s | 0.25 | 0.6 | 2 |
| Pings/ensemble | | 4 | 20 | 300 |
| Ensemble interval | minutes | 0.08 | 1 | 30 |
| Velocity uncertainty ($\sigma$) | cm s$^{-1}$ | 2.5 | 2.1 | 2.1 |

## 3.2 Glider instrumentation

The ALSEAMAR SeaExplorer glider was equipped with a 1 MHz Nortek AD2CP Acoustic Doppler Current Profiler with a downward-looking configuration in the same way as in de Fommervault et al. (2019). The glider was programmed to dive with a pitch angle of approximately ±22°, with the design of the AD2CP allowing 3 beams in a optimal 'Janus' configuration (Mullison et al., 2013) either for descents or ascents (Ma et al., 2019). The AD2CP compass is calibrated far from any magnetic disturbance to avoid biasing the current direction. In the present study, the calibration has been realized with the AD2CP mounted on the fully equipped glider, using the standard ALSEAMAR and Nortek (2022) procedures.

The AD2CP was configured to collect measurements in beam coordinates in 15 sampling cells, with a 2 m resolution and a 0.2 m blanking distance (Table 2). The profiler pinged 4 profiles per 1-second average ensemble, every 5 s. Bottom tracking was configured at the same sampling frequency, with a 0.1 m blanking distance, to detect the bottom position in the lowest ten meters of the water column. Velocities were acquired in beam coordinates.

To measure optical turbidity throughout the water column, the dry payload section of the SeaExplorer was also equipped with a Seabird WetLabs ECO-Puck FLBBCD sensor, which notably delivered light scattering measurements (expressed in m sr$^{-1}$) at a wavelength of 700 nm. A Seabird Glider-Pumped Conductivity, Pressure and Temperature probe (GPCTD), was also integrated into the nose of the SeaExplorer, to acquire ancillary hydrological measurements. The FLBBCD probe acquired optical backscattering every 1 s, and the GPCTD probe measured every 4 s.



### 3.3 Bottom moorings instrumentation

Each mooring contained an upward-looking RD Instrument ADCP, both operating at 300 kHz, each of them having four transducers forming 4 beams equally slanted with a 20° angle from the vertical direction. Table 2 specifies their configurations.
Their vertical resolutions and ranges were identical but their temporal resolutions differed. CIAM ADCP was configured to acquire data with high temporal resolution as it had been specifically deployed for the MELANGE validation campaign. The temporal resolution of GV1 was lower (10 min measurements, averaged every 30 min) but it still allowed to collect data at a relevant timescale given the tidal periodicity of about 12 hours. Compass calibration was performed on both ADCPs following the Le Menn and Pacaud (2015) procedure.

### 3.4 CTD-Rosette instrumentation

The CTD-Rosette sampler was notably equipped with a Seabird SBE911plus CTD and a WetLabs FLBBCD probe (same model as the glider), as well as being equipped with a Sequoia Scientific particle size analyzer (LISST-100X Type C) to measure volume distribution of suspended sediment in 32 size classes, logarithmically distributed from 2 to 380 $\mu$m (Agrawal and Pottsmith, 2000). Water samples were collected from the CTD-Rosette for analysis of suspended sediment concentration
in particular.

## 4 Methods

The processing method used to recover the absolute currents recorded by an ADCP onboard a glider has been chosen as the well known "LADCP shear method" (Visbeck, 2002; Ordonez et al., 2012) presented in subsection 4.1, with computation of barotropic component in subsection 4.2 and fluxes in subsection 4.3. Concerning SPMC, an ADCP is also able to recover
an index of turbidity from the acoustic measurement following a method presented in subsection 4.4. The assessment of the velocity uncertainties is given in subsection 4.5, while for the moorings it is given in subsection 4.6. Final errors given later in the results are computed according to metrics specified in subsection 4.7. Contrary to the currents, the acoustic and even the optical measurements of the particle concentrations need in situ calibrations. They are presented in subsections 4.8 and 4.9.

### 4.1 Obtaining ocean currents with a glider-mounted ADCP

Processing of glider-AD2CP data is typically done at the end of a mission, once the full resolution dataset has been recovered. In the following section we describe the main steps of the Delayed Mode algorithm. The AD2CP measurements take into consideration two contributions: the water current and the motion of the glider relative to the water body. To obtain the ocean contribution as a full water column profile, for each single yo of the glider, the LADCP shear method (Visbeck, 2002) was used. Assuming that the velocity of the glider is constant during a ping ($\approx$10 ms), and that the ocean current is stationary during
a yo ($\approx$20 min), the method is applicable to the AD2CP measurements $U_{\text{AD2CP}}(z)$, such that:



$$U_{\text{AD2CP}}(z) = (U_{\text{barotropic}} + U_{\text{baroclinic}}(z)) + U_{\text{GLI}}(z) \tag{1}$$

where $U_{\text{GLI}}$ is the glider velocity relative to the water mass, $U_{\text{barotropic}}$ and $U_{\text{baroclinic}}$ are the depth-independent and depth-dependent ocean current components, respectively, so that the total ocean current is $U_{\text{ocean}}(z) = U_{\text{barotropic}} + U_{\text{baroclinic}}(z)$. Based on Eq. (1), and treating $u$ (east-west) and $v$ (north-south) components separately, we then: i) computed elementary shear profiles using central differences; ii) averaged overlapping shears; and iii) retrieved ocean currents by integrating shears, using the ocean velocities from the bottom track as the integration constant (Gentil et al., 2020). As the bottom track shows no variability with respect to sediment type or glider heading, it has been selected as a reference for constraining the relative velocity profiles.

Quality controlled tests are detailed in de Fommervault et al. (2019). The main pipeline of operations is set up:

1. to correct the speed of sound using the salinity and temperature measured by the glider;

2. to discard data with less than 50% correlations or less than 3 dB signal-to-noise ratios;

3. to discard data when the glider is pitched or rolled enough to displaced cells from their nominal alignment (threshold of 20° around the nominal position);

4. to discard data out of the water column;

5. to linearly extrapolate missing values close to the sea surface;

6. to discard outliers for which vertical shear velocity exceeds 0.02 s⁻¹;

7. to discard data farther away from the ADCP that are not consistent with those coming from profiles in close vicinity (Todd et al., 2011);

8. to discard data where vertical velocities in two adjacent cells during a ping exceeds the threshold of 0.05 m s⁻¹ (Visbeck, 2002);

9. to discard profiles with too many missing or extrapolated values;

10. to compute the mean shear in each depth interval only where more than 5 remaining (good) values are available.

The final measurements obtained were given as profiles with 2 m cells.

### 4.2 Computing barotropic velocities

The semi-diurnal tide is the main hydrodynamic forcing on the MELANGE area (Vincent and Le Provost, 1988). Accordingly, the measured ocean current was analysed from two perspectives: i) the total current, that is, the current varying with both depth (cell) and time (profile), and ii) the barotropic current, computed as the depth-averaged current, which we approximated as the tidal current due to being mostly driven by tide. The bottom boundary layer was estimated to be approximately 9 to 14 m thick during the 5-day campaign, using the Soulsby (1983) formula for tidal flows. Depth varied along the trajectory of the glider from -121 m to -105 m, due to bathymetric variations and tidal range. In order to compare a consistent barotropic layer of the water column for the three current profilers, given their respective deployment depth, data availability near the surface, and





time-varying excursion of the tidal boundary layer from the seafloor, the depth-average was calculated for $u$ and $v$ separately as:

$$u_{\mathrm{t}}(t) = \int_{-98 \text{ m}}^{-52 \text{ m}} u(t, z)\,dz \tag{2}$$

with $t$ the time of the glider-AD2CP profile and $z$ the depth computed from the free surface. In the submesoscale MELANGE area, the mesoscale tidal range is considered spatially homogeneous, as it operates on a larger scale than the studied area. Therefore, the depths of the current profiles are given hereafter with respect to the tide-varying free surface position, rather than with respect to the space and time-varying bathymetry.

### 4.3    Computing suspended sediment fluxes

The instantaneous flux of SPM, at all depths and locations of geographic points surveyed, notated $Q_{\mathrm{F}}$, and the flux integrated in depth, notated $Q_{\mathrm{S}}$, with their two components $u$ and $v$, are calculated separately as follows:

$$Q_{\mathrm{S}u}(t) = \int_{z(\text{bottom})}^{z(\text{surface})} Q_{\mathrm{F}u}(t, z)\,dz = \int_{z(\text{bottom})}^{z(\text{surface})} SPMC(t, z) \cdot u(t, z)\,dz \tag{3}$$

where $t$ the time of the glider-AD2CP profile, $u$ and $v$ the two velocity components and $z$ the depth from the bottom ocean to its surface. The missing data $(u,v)$ and $SPMC$ located near the bottom and the surface are completed by extrapolation (of 205 log-transformed values for $SPMC$).

### 4.4    Obtaining acoustic backscatter from ADCP data

Acoustic backscatter values were also obtained from the glider-AD2CP profiler and provided an additional (acoustic related) proxy of the suspended sediment concentration (Thorne and Hanes, 2002). The acoustic backscatter $S_{\mathrm{v}}$ [dB], also referred to as volume backscattering strength in the literature, is a metric of the acoustic signal returned by the scatterers in a finite 210 cell-size volume. As in optics, the acoustic backscatter is linked to particulate abundance, and therefore to concentration, but also to particle size as well as to shape and mechanical contrasts with the surrounding fluid, see e.g. Stanton (1989); Pieper and Holliday (1984). However, optical backscattering is highly sensitive to fine particles (typically less than 30 $\mu$m) while acoustic backscattering, especially at 1 MHz, is sensitive to coarser particles having a typical size diameter of about 1 mm (signal related to the fourth power of the grain radius (Stanton et al., 1998)). In the present work, $S_{\mathrm{v}}$ [dB] is computed from the 215 echo amplitude $E$ [counts], i.e. the raw output values given by the AD2CP, as follows (Jourdin et al., 2014; Nortek, 2022):



$$S_\text{v} = RL + 2TL + DT - SL - 10\log_{10}(V)$$
$$RL = 10\log_{10}\left(10^{K_\text{c}\cdot(E-E_0)/10} - 1\right)$$
$$TL = 20\log_{10}(\psi R) + \alpha_\text{w}R$$
$$V = \phi \cdot (\psi R)^2 \cdot \frac{c\tau}{2}$$

(4)

The reverberation level $RL$ is calculated according to the formula by Mullison (2017), after Gostiaux and van Haren (2010), with $E_0$ [counts] the echo noise floor and $K_\text{c}$ [dB/count] the conversion slope. Transmission loss $TL$ is calculated using the range of acoustic cells from transducers $R$, associated with near-field correction $\psi$ (Downing et al., 1995) and the coefficient of

sound absorption $\alpha_\text{w}$ [dB m$^{-1}$]. Our calculation takes into account the contribution of water only, through the formula given by Francois and Garrison (1982): we consider the absorption by suspended particles to be negligible given their low concentrations (cf. Section 5.2.1) far less than 100 mg l$^{-1}$ (Tessier et al., 2008). $SL$ and $DT$ [dB] represent the Sound Level emitted by the instrument, and its Detection Threshold, respectively set to 217 dB and 100 dB (Nortek AS, pers. commun.). The velocity of sound in water $c$ [m s$^{-1}$] is set to 1500 m s$^{-1}$. The volume of an acoustic cell $V$ can be estimated with $\tau$ [s], the pulse duration

of the acoustic signal, equal to 2.94 ms for our instrument and $\phi$ the solid beam angle [sr] corresponding to an aperture angle of 2.9°.

In the following analysis, acoustic data recorded in near-surface layers of the ocean (from 0 to 20 m depth) were discarded due to the very noisy accompanying backscatter signal. This is most likely the result of contamination by air bubble clouds which act as high-strength scatterers (Van Haren, 2001). The presence of air bubbles, which are created at the air-water interface

and advected downward, is related to rough wind and wave conditions (Wang et al., 2011; Vagle et al., 2010), that occurred during the present campaign (cf. Figure . 2).

### 4.5 Uncertainty assessment of glider AD2CP velocities

To validate the current estimates, the AD2CP measurement uncertainty for the reconstructed profiles has to be considered. For the Nortek 1 MHz instrument integrated on the SeaExplorer and cells with a resolution of 2 m, the manufacturer specifies a

typical ping uncertainty of approximately 6 cm s$^{-1}$ (the same as for the bottom track ping). With 4-ping ensembles per second (Table 2), the precision of a single ensemble beam velocity becomes 3 cm s$^{-1}$. Velocity data were recorded every 5 s with an average glider vertical velocity of 0.2 m s$^{-1}$, giving a number of overlapping data ranging between 1 for the first bin at the surface to 30 on average over most of the profile. By discarding cells with less than 3 overlaps and propagating errors, both due to water column and bottom track pings, a typical standard deviation error $\sigma(\text{GLI}) = 2.5$ cm s$^{-1}$ for both components of

the horizontal velocities has been estimated.

### 4.6 Uncertainty assessment of moored ADCP velocities

First, a quality-control filtering was performed, based on standard recommendations (Gordon and RDI, 1996): cells with i) a range above the maximum range predicted by PlanADCP, ii) a beam correlation lower than 64 counts, iii) a sum of the percents





good of beams 1 and 4 lower than 75%, and iv) an absolute error velocity higher than 6 cm s$^{-1}$, were discarded. For the CIAM
data, the profiles, available every minute, were averaged over the glider profile's inherent averaging period. This averaging
also allows computation of corresponding standard deviations for the CIAM horizontal velocities. Hence, a typical standard
deviation error $\sigma(\text{CIAM}) = 2.1$ cm s$^{-1}$ for both components of the horizontal velocities has been estimated, knowing that this
standard deviation encompasses not only measurement errors but also the ocean variability at the averaging scales used, over
the vertical and in time with $O(30\text{min})$. For the GV1 data (10 min of measurements averaged every 30 min), we assumed a
similar standard deviation value of $\sigma(\text{GV1}) = 2.1$ cm s$^{-1}$ knowing that the two ADCPs were similarly configured and moored
in the same area, with averaging timescales of $O(10 - 20\text{min})$ for GV1.

### 4.7  Further error metrics

ADCP data from the GV1 and CIAM moorings were post-processed in order to match the AD2CP data characteristics of
the glider acquisitions. ADCP-AD2CP vertical cells were matched when their centres were less than 1 m away. The vertical
positions of the cells were then expressed with respect to the free surface (which is the pressure reference for the glider), using
the ADCPs' integrated pressure sensors. Finally, glider and mooring profiles were temporally matched. As said in the previous
subsection, CIAM data were averaged over the glider profile's inherent averaging period. For the GV1 data, the closest profiles
to the glider profiles within a 30 min interval were used. Cells with less than 20% of overlaps throughout the averaging period,
due to failed quality controls, were discarded.

For current comparisons, Root Mean Squared Differences (RMSD) are computed as follows:

$$\text{RMSD} = \text{RMSD}(X) = \sqrt{\frac{1}{N}\sum_{i=1}^{N}\left(X(\text{Platform2})_i - X(\text{Platform1})_i\right)^2} \tag{5}$$

where $X$ is one velocity component; Platform1 and Platform2 can be either GLI, CIAM or GV1; $i$ indicates a common bin
cell and $N$ is the total number of bin cells considered in the comparison.

In order to compare the RMSD obtained (in Section 5.1.1) with the a priori error estimations performed in previous Sec-
tions 4.5 and 4.6, a Combined Uncertainty (CU) has also been defined as follows:

$$\text{CU} = \sqrt{\sigma^2(\text{Platform1}) + \sigma^2(\text{Platform2})} \tag{6}$$

where $\sigma$ is the standard deviation that typically is worth 2.5 cm s$^{-1}$ for the glider and 2.1 cm s$^{-1}$ for the moorings. With this
definition of CU we obviously assume that errors coming from different platforms are uncorrelated.

Concerning suspended matter, the Root Mean Squared Log Differences (RMSLD) and Relative Percent Difference (RPD)
are used:

$$\text{RMSLD} = \text{RMSLD}(Y) = \sqrt{\frac{1}{M}\sum_{j=1}^{M}\left(\log_{10}\left(Y(\text{System2})_j\right) - \log_{10}\left(Y(\text{System1})_j\right)\right)^2} \tag{7}$$



$$\text{RPD} = \text{RPD}(Y) = 100 \cdot \frac{1}{M} \sum_{j=1}^{M} \frac{|Y(\text{System2})_j - Y(\text{System1})_j|}{Y(\text{System1})_j} \tag{8}$$

where $Y$ is SPMC or a backscattering intensity; System1 and System2 are measurement systems such as filter weighing, optical or acoustic backscattering, where System1 is considered as a reference; $j$ identifies one pair of compared measurements, with $M$ being their total number.

### 4.8 In situ SPMC calibration of the optical sensors

The glider-mounted WetLabs BB700 optical backscattering sensor outputs were converted from volume scattering measurements into optical particulate backscattering coefficients $b_{\text{bp}}$ [m⁻¹] as proposed by Schmechtig et al. (2018); Boss and Pegau (2001), using the temperature and salinity concomitantly measured by the GPCTD of the glider to remove the water backscattering contribution (Zhang et al., 2009). The outputs of the CTD-Rosette-mounted WetLabs BB700 optical backscattering sensor (same sensor model and wavelength) were processed identically.

In order to obtain a direct ground estimation of the SPMC (in [mg l⁻¹]), the water samples collected with the CTD-Rosette sampler were filtered, and the corresponding filters were weighted following a standard procedure based on triplicates, e.g. Neukermans et al. (2012). SPMC measurements from filtrations were used to calibrate the CTD-Rosette optical scattering acquisitions, in order to retrieve SPMC from the latter. To do so, the optical measurements acquired when the Niskin bottles were closed (6 measurements) were averaged. The corresponding means were linearly fitted with SPMC values obtained from filtration, taking the $\log_{10}$ of the two parameters to perform the fit in order to improve the statistical estimation (Figure. 3).

The calibration equation was taken for the best coefficient of determination ($R^2 = 0.91$), obtained when combining the 14 and 18 February 2021 datasets for the bottom measurements, and also leads to the following RMSLD and RPD error values (filter weighing being the reference):

$$\begin{aligned}
\log_{10}(SPMC) &= 0.8 \times \log_{10}(b_{\text{bp}}) + 1.8 \\
R^2 &= 0.91 \\
\text{RMSLD}(SPMC) &= 0.11 \log_{10}(\text{mg} \cdot \text{l}^{-1}) \\
\text{RPD}(SPMC) &= 17\%
\end{aligned} \tag{9}$$

Bottom data giving the best fit is consistent with numerous observations reported in the literature, as seen for example in Fettweis et al. (2019). Combining the 14-18 February datasets has the advantage of providing a calibration equation that is valid over the full campaign duration.

As the Glider and CTD-Rosette BB sensors were identical, and considering each of the sensors truly measure optical backscattering, the SPMC to $b_{bp}$ calibration of the CTD-Rosette will be applied to the $b_{bp}$ measurements of the Glider (next subsection).



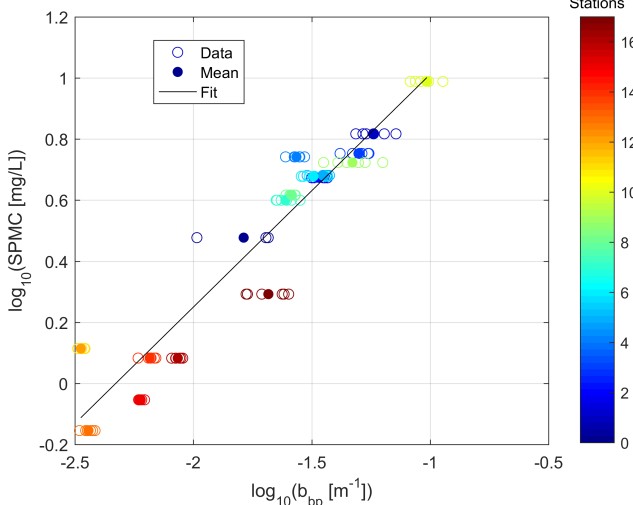

**Figure 3.** Calibration between the optical particulate backscattering coefficients $b_{bp}$, and the Suspended Particulate Matter Concentrations (SPMC), measured with the CTD-Rosette on 17 stations (see locations in Fig. 1). Data correspond to the bottom measurements, with three to six $b_{bp}$ values for one SPMC value; Mean is the mean $b_{bp}$ value on this three to six $b_{bp}$ measurements; Fit is the obtained linear regression, spelled out in Equation 9.

### 4.9 In situ SPMC calibration of the glider-AD2CP

Figure 4 shows the correlation between optical and acoustic backscattering coefficients in the deep layer. Such correlation
is expected for suspended sediment particles, especially in winter (Tessier et al., 2008). This allows to find the following representative ($S_v$, $b_{bp}$) relationship for the full MELANGE period, after linear regression, and also leads to the following RMSLD and RPD error values (optical backscattering being the reference):

$$\log_{10}(b_{bp,Gli}) = 0.07 \times S_v + 0.2$$
$$R^2 = 0.88$$
$$\text{RMSLD}(b_{bp}) = 0.09 \log_{10}(\text{m}^{-1})$$
$$\text{RPD}(b_{bp}) = 15\%$$

(10)

Doing so, we intentionally ignore the surface layer because micro-bubbles and living particles (plankton) can perturb the
linear regression because their acoustic versus optical response can be extremely variable (Jourdin et al., 2014). Corresponding error values in terms of SPMC for Equation (10) can be computed: if we apply the factor 0.8 of Equation (9) to error values for $b_{bp}$, then we obtain RMSLD($SPMC$)=0.07 $\log_{10}(\text{mg} \cdot \text{l}^{-1})$ and RPD($SPMC$)=12%. Those error values are about 30% lower than those obtained with the calibration Equation (9).





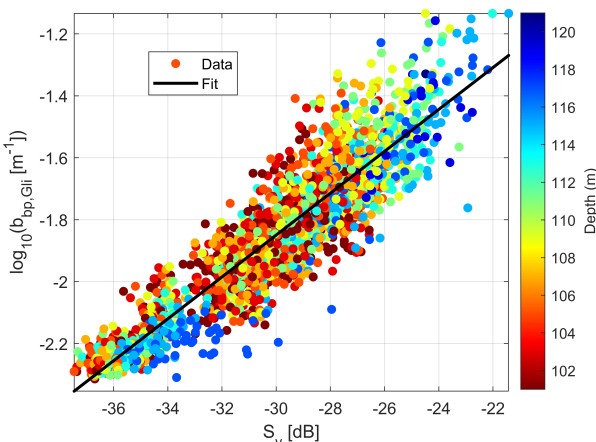

**Figure 4.** Calibration between the optical particulate backscatter coefficients $b_{bp}$ and the acoustic backscatter of the AD2CP $S_v$ measured by the glider in the deep layer. Data correspond to the glider measurements ($N = 2547$). Fit is the obtained linear regression, spelled out in Equation 10.

We propose here to calibrate the turbidimeter (backscattering sensor) that is onboard the glider using the same calibration
equation (Equation 9) obtained for the turbidimeter onboard the CTD-Rosette. Such an implementation is valid only if both the glider's and the CTD-Rosette's optical turbidimeters operate identically (Fettweis et al., 2019). They are the same make and model and perform at the same wavelength. Combining this equation with the previous Equation 10 leads to an estimation of the SPMC from the glider $S_v$ according to the following new equation:

$$\log_{10}(SPMC) = 0.06 \times S_v + 2.0 \tag{11}$$

**5  Results & Discussion**

In assessing suspended particulate transport and quantifying hydro-sedimentary processes (presented in Subsection 5.3), the glider needs the ability to sufficiently measure both the general currents (presented in Subsection 5.1) and the suspended particulate concentrations (presented in Subsection 5.2).

**5.1  Validation of Glider currents**

Here the total currents observed are first presented before we interpret the results according to their two hydrodynamic components: barotropic and baroclinic.



### 5.1.1 Total current

Figure 5 shows that the two current components observed by all platforms display similar patterns and intensities, with a consistent tidal current periodic signal (with M2 the main tidal component with a period of 12h25'). This comparison relies on the assumption that the spatial variability of the ocean current, at the scale of the MELANGE area, is not significant compared to its temporal variability, under the conditions encountered with the tidal forcing. This assumption could be tested by first comparing the measurements of the two moorings (Figure. 6a). A linear regression analysis performed on $N \approx 8000$ matchup cells gave determination coefficients $R^2$ higher than 0.95 for the two components (Table 3) with a highly significant p-value (<0.001), which demonstrates the relevance of this assumption. In terms of current intensities, the root-mean square difference (RMSD) between the components of the two moorings ranges from 2.4 to 3.1 cm s$^{-1}$ (which corresponds to about 10% of the maximum value of 0.3 m s$^{-1}$). This is also consistent with the Combined Uncertainty (CU) of $O(3$ cm s$^{-1})$ between CIAM and GV1 (Table 3). Thus the intensity of the ocean current is consistently measured by the two platforms across the study area during the validation campaign.

Moreover, the least-square linear regression analyses indicate a good agreement between the glider and each of the two moorings (Figure 6; Table 3). Determination coefficients $R^2$ range mostly from 0.93 to 0.97 with a significant p-value (<0.001), nearly as good as those between the two-moorings. The RMSD of about 3 cm s$^{-1}$ between GLI and CIAM are very satisfactory and consistent with the Combined Uncertainty. Yet this is not exactly the case for the comparison between GLI and GV1 with a RMSD of about 4 cm s$^{-1}$. This overestimation of uncertainty might be attributed to the proximity of GV1 to the shipwreck Erika (long term moorings are usually put near shipwrecks in order to avoid bottom trawling), potentially causing a minor deviation of its compass. Indeed intercepts $b$ derived in Table 3) show values round +1 and -1 cm s$^{-1}$ for the comparison of GV1 with other platforms. This suggests a possible systematic bias (of the compass) in the GV1 measurements which can also explain the larger values of RMSD between GLI and GV1.

Values of slopes $a$ in Table 3 show that the glider slightly underestimates the current intensities. This underestimation is still of a lower order of magnitude than the studied forcing. If we presume that the measuring errors of CIAM and GV1 are fully accounted for, this glider underestimation might come from the processing of the glider data. The shear method was applied on single-yos, resulting in the average of overlapping data up to 25 min, providing they had the same vertical position. This timescale may represent the limit of relevancy for considering a steady tidal current, especially during spring tide. Furthermore, the glider moved horizontally while diving down and up, ending its single yo trajectory about 500 m away from where it started. Therefore, data collected on the up-cast may be slightly different from those of the down-cast.

The results match the values reported in the literature with in situ validations for a wide variety of well-documented deployment and settings of gliders (Table 4). In particular, the mean RMSE of comparison between GLI (glider) and CIAM (mooring) round 3 cm s$^{-1}$ is in agreement with those obtained in studies also using bottom track referencing, an approach yielding to the lowest uncertainties compared e.g. to dive-averaged referencing (Thurnherr, 2010; Ordonez et al., 2012), either obtained for an optimized angle of attack (Ellis et al., 2015), or for the ideal case of sufficient scatterers in suspension (Thurnherr, 2010).





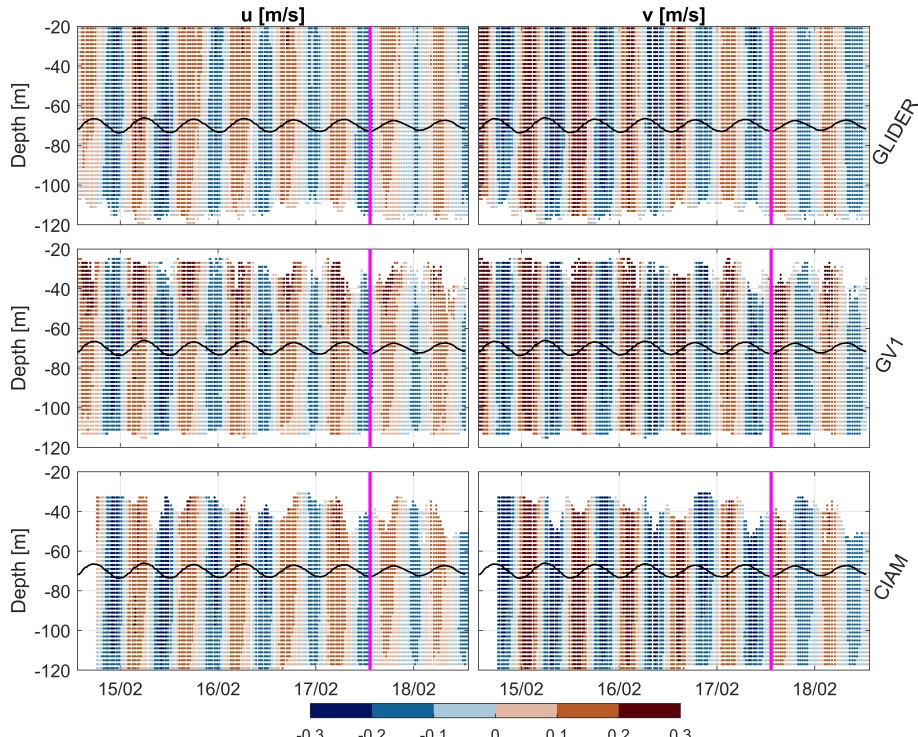

**Figure 5.** Current components measured by the glider-AD2CP and moored ADCPs from GV1 and CIAM. Glider missing data near the bottom result from reversing of the glider trajectory near the seafloor. GV1 missing data near the bottom is due to the higher position of the ADCP relative to the seafloor compared to CIAM. Moorings' missing data close to the surface correspond to the maximum detection range of the ADCPs. The black line represents the free surface variation with tide, computed from the GV1 pressure sensor. The vertical magenta line delineates the BU and VM survey periods.

### 5.1.2 Barotropic current

Tide is the main forcing driving currents in the glider validation area as displayed in Figure 5, so we focus herein on the barotropic current (here computed according to section 4.2). Figure 7 shows that these components range from -0.2 to 0.2 m s$^{-1}$ for $u_\mathrm{t}$, and -0.3 to 0.3 m s$^{-1}$ for $v_\mathrm{t}$. In terms of magnitude, all three current profilers show ratios of barotropic current over total current equal to 1 on average. In terms of variations, the whole STD (standard deviation) for the current components is around 0.06 m s$^{-1}$ (in the barotropic layer). For a given mooring profile, the STD is in the order of $O(1 \text{ cm s}^{-1})$, and for a given glider profile, the STD is in the order of $O(2 \text{ cm s}^{-1})$.

As expected from the previous analyses on total current, comparisons yield similar, very satisfying results. The barotropic current estimated from the two moorings are in good agreement: $R^2$ are comprised between 0.97 and 0.99 and RMSD are in the order of $O(2 \text{ cm s}^{-1})$ (which corresponds to about 10% of the typical maximum value) for all current components and survey periods considered. The glider barotropic components also show good agreement with those of the moorings. Determination



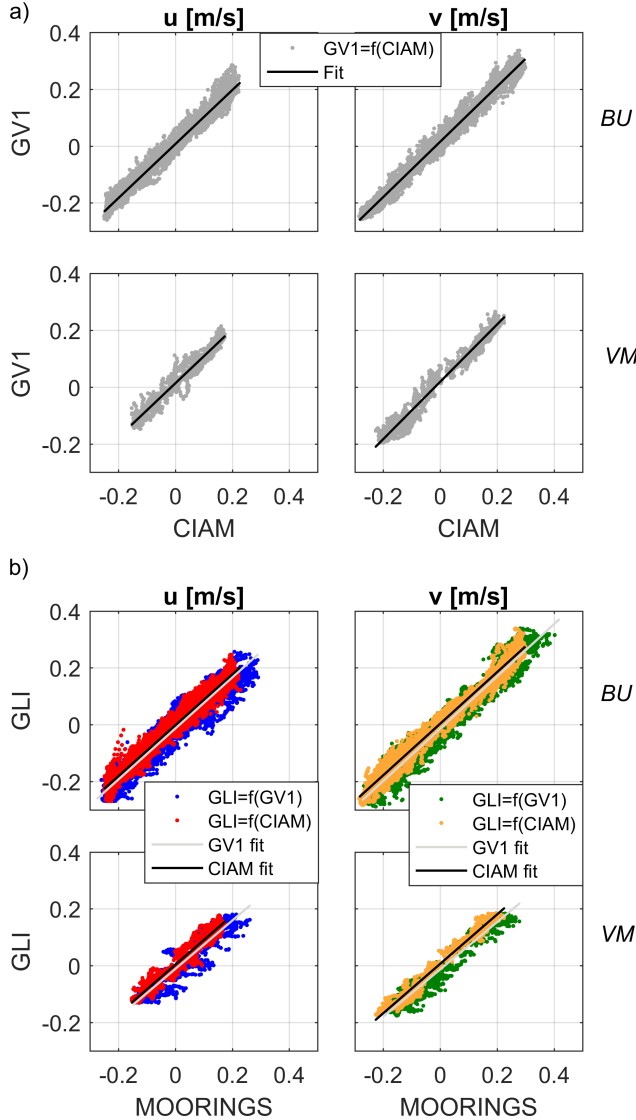

**Figure 6.** Total current least-square regressions for $u$ and $v$ current components: **(a)** CIAM versus GV1 moorings comparison; **(b)** Glider versus CIAM and GV1 moorings comparison. Subplots distinguish BU and VM data. Associated regression coefficients are given in Table 3.

coefficients are comprised between 0.93 and 0.98, which is slightly better than for the total current comparison. Least-squares slopes and intercept are similar, and RMSD are also $O(2 \text{ cm s}^{-1})$.

The highest differences are found around each component maxima (see Figure 7), i.e. 2 h before high tide for $u_t$ and 4 h before high tide for $v_t$, and are up to 6 cm s$^{-1}$ for both components. In particular, from that Figure, we can observe some underestimations of the currents by the glider, probably for the same reasons as stated in the last paragraph of the previous section 5.1.1.



**Table 3.** Comparison of ADCP horizontal velocity components ($u$ and $v$) recorded from two different platforms. Indexes BU and VM respectively stand for the BUtterfly-pattern and Virtual-Mooring survey periods. Slopes $a$, intercepts $b$ (cm s$^{-1}$) and corresponding determination coefficients $R^2$ are obtained after linear regression between the velocity components from the two compared platforms, also with RMSD (cm s$^{-1}$) from Eq. 5 and Combined Uncertainties (CU) from Eq. 6. All regression p-values are $< 0.001$, meaning the affine-linear relationship is significant. Nota bene: CU values varies slightly depending on the considered component and period of comparison.

| Platform1 | Platform2 | Curr. | $a$ | $b$ | $R^2$ | RMSD | CU |
|---|---|---|---|---|---|---|---|
| CIAM | GV1 | $u_{\mathrm{BU}}$ | 0.95 | 0.7 | 0.98 | 2.4 | 3.0 |
| | | $v_{\mathrm{BU}}$ | 0.97 | 1.5 | 0.98 | 2.7 | 2.7 |
| | | $u_{\mathrm{VM}}$ | 0.95 | 1.4 | 0.95 | 2.5 | 2.8 |
| | | $v_{\mathrm{VM}}$ | 1.01 | 1.9 | 0.97 | 3.1 | 2.7 |
| GV1 | GLI | $u_{\mathrm{BU}}$ | 0.90 | -1.3 | 0.94 | 3.9 | 3.2 |
| | | $v_{\mathrm{BU}}$ | 0.92 | -1.4 | 0.94 | 3.7 | 3.1 |
| | | $u_{\mathrm{VM}}$ | 0.85 | -1.1 | 0.89 | 3.7 | 3.2 |
| | | $v_{\mathrm{VM}}$ | 0.83 | -1.3 | 0.95 | 3.7 | 3.1 |
| CIAM | GLI | $u_{\mathrm{BU}}$ | 0.90 | -0.2 | 0.96 | 3.0 | 3.2 |
| | | $v_{\mathrm{BU}}$ | 0.91 | 0.3 | 0.97 | 3.0 | 3.1 |
| | | $u_{\mathrm{VM}}$ | 0.87 | 0.4 | 0.93 | 2.7 | 3.2 |
| | | $v_{\mathrm{VM}}$ | 0.87 | 0.7 | 0.97 | 2.6 | 3.1 |

### 5.1.3 Baroclinic current

The baroclinic current $u_c(T, z)$ is computed as the difference between total and barotropic currents: $u(T, z) - u_t(T)$. Baroclinic currents show similar patterns between the two bottom-moored ADCPs (not shown here); therefore, only the comparison between glider and CIAM data, which offers the highest temporal resolution, is discussed here.

Baroclinic current for $u$ and $v$ components approximately range from $\pm 0.15$ cm s$^{-1}$ to $\pm 0.1$ cm s$^{-1}$, for the glider and CIAM platforms respectively. In terms of magnitude, their averages are 0.03 cm s$^{-1}$ and 0.02 cm s$^{-1}$, respectively. The same values are found for the STD. Differences between the glider and CIAM estimations are mostly comprised in [-0.02; 0.02] cm s$^{-1}$ (Figure 8). This consistency highlights the ability of glider to accurately capture the spatial variability of baroclinic currents in this environment. However, maxima differences are found near the seafloor and the surface (Figure 8). These regions coincide with the glider's dive and apogee phases, where the number of overlaps used to reconstruct the profile is lowest (around three after quality control). This low overlap count leads to a higher dispersion of the data, which likely explains the discrepancies observed between the two platforms.

In addition to confirming total current measurements, this study demonstrates for the first time that the decomposition of currents into barotropic and baroclinic components also yields reliable results. The validation of these components, with strong



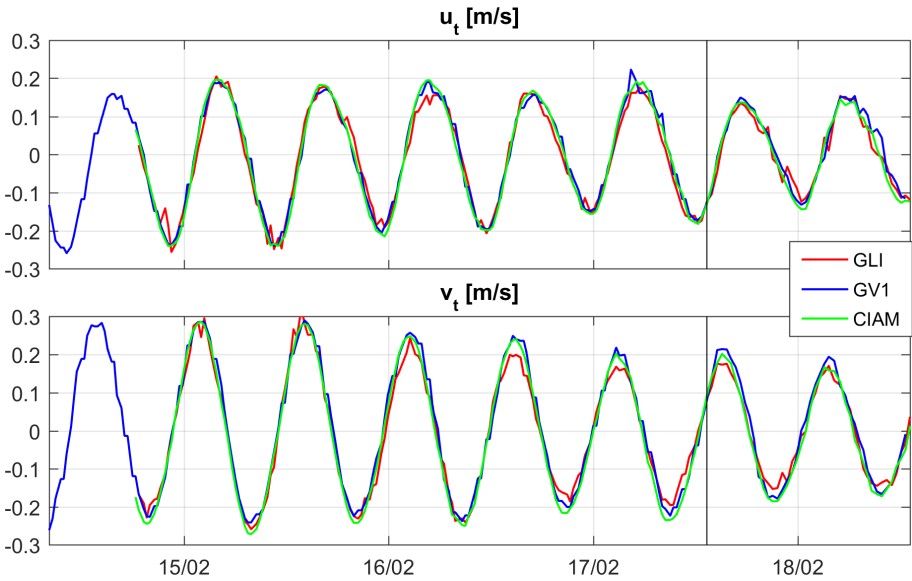

**Figure 7.** Barotropic current components computed from equation (2), for the glider-AD2CP (GLI) and moored-ADCPs (CIAM, GV1). The vertical black line delineates the BU and VM survey periods.

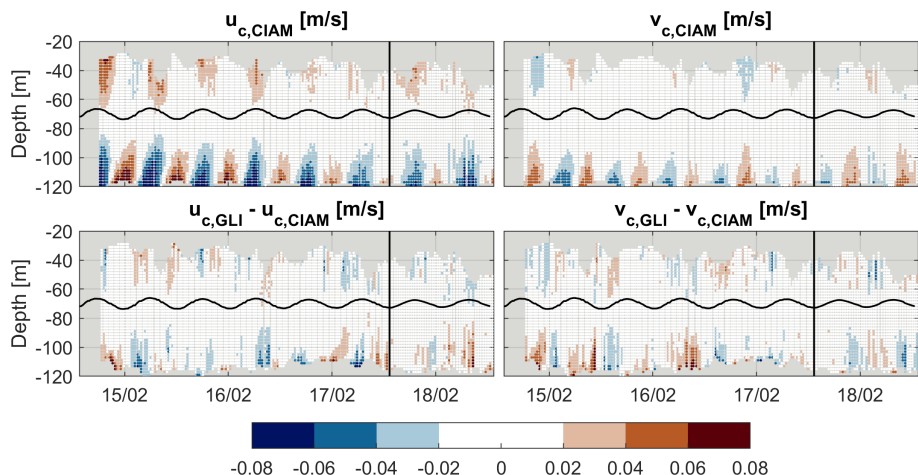

**Figure 8.** Baroclinic current comparison: up) Components of the CIAM moored-ADCP; down) Difference between the components of the glider-AD2CP and CIAM moored-ADCP. Baroclinic components were computed according to equation (2).The black undulated line represents the free surface variation with tide, computed from the GV1 pressure sensor. The black straight line delineates the BU and VM survey periods.

agreement between glider and mooring data, highlights the glider's capability to capture not only total currents but also their underlying hydrodynamic structures.





## 5.2 SPMC Estimates

Subsection 5.2.2 gives a description of the spatio-temporal dynamics of the turbidity observed by the glider. But, first, sub-
390 section 5.2.1 compares the optical and acoustic sensing of turbidity in terms of Suspended Particulate Matter Concentration
(SPMC).

### 5.2.1 Optical and acoustic comparison

In assessing SPMC, the main sensor to put on board a glider obviously is a turbidimeter (Maa et al., 1992), provided that its
output units, either in FNU, NTU or in m$^{-1}$, is well converted to mg l$^{-1}$ according to a proper *in situ* calibration (as shown
in Section 4.8). However turbidimeters are not always well correlated to SPMC (Downing, 2006). A more comprehensive
observation of SPM can make use of both optical and acoustic sensors (Fettweis et al., 2019). For example, Haalboom et al.
(2021) use a turbidimeter, a transmissometer, and also an ADCP to get a better picture of the types and concentrations of SPM.

Table 5 provides statistics for SPMC values measured using the two optical FLBBCD sensors (one onboard the CTD-
Rosette and the other onboard the glider) and the acoustic AD2CP sensor (onboard the glider). When applied to the whole
survey period, these statistics show fairly consistent SPMC ranges between all sensors, with in particular median values round
1.2 mg l$^{-1}$ and interquartile range (IQR) values round 0.5 mg l$^{-1}$. The maximum acoustic-derived SPMC of 4.7 mg l$^{-1}$ and
optical-derived SPMC of 7.8 mg l$^{-1}$ are recorded near the seafloor on 1200 UTC 14 February.

At the specific dates of 14 and 18 February (where the research vessel and glider closer than 6 NM each other), median
SPMC values derived from both optical sensors are almost identical (by 0.1 mg l$^{-1}$), but interquartile range (IQR) values show
that the CTD-Rosette profiles appear to be more widely spread. Also, at these two dates, acoustic SPMC median values appear
to be lower than the optical ones by roughly a factor two.

### 5.2.2 Temporal and vertical dynamics

Figure 9 displays the temporal evolution of SPMC simultaneously recorded by the glider in three vertical layers: Deep, Middle
and Surface. Since the acoustic and optical backscattering coefficients derived from its AD2CP and FLBBCD sensors are cor-
410 related (see Figure 4), both acoustic and optical-derived SPMC obviously show similar patterns. However, some differences are
observed, with the optical sensor recording higher SPMC near the seafloor compared to the acoustic sensor. This discrepancy
suggests that the bottom layer is primarily composed of fine particles, which are more effectively detected by the optical sensor
than by the acoustic one.

In terms of temporal evolution, the glider observes a higher vertical dynamics during the BU period than during the VM
period. In particular two intense events appear in the BU period. They are centered around 1700 UTC 14 February, and around
0000 UTC 17 February. During these first and second events, optical values of SPMC in the deep layer are up to 7.8 mg l$^{-1}$
and 5 mg l$^{-1}$, respectively, indicating the presence of a well-developed bottom nepheloid layer with a thickness of 15 to 20m
(Figure 9). In the intermediate and surface layers both acoustic and optical SPMC are consistent and give values mostly





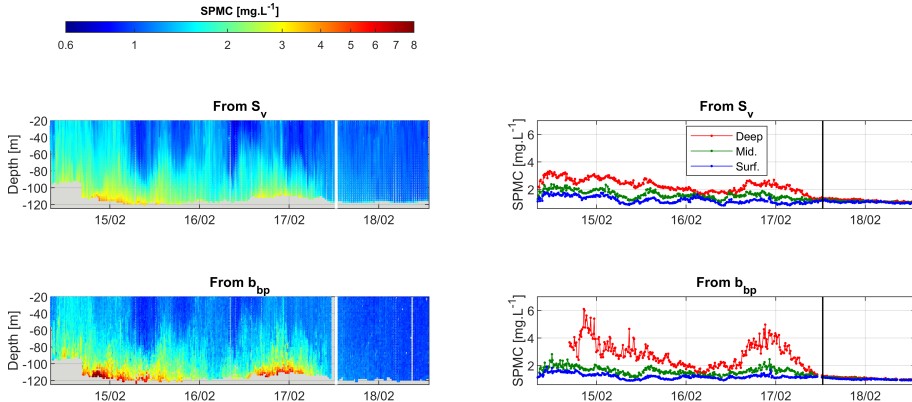

**Figure 9.** SPM concentrations recorded by the glider. Top: SPMC from acoustic backscattering ($S_v$). Bottom: SPMC from optical backscattering ($b_{bp}$). On the left are hovmöller diagrams. On the right are solid lines displaying corresponding levels in the three main water column layers defined by the following depth ranges: Deep=[-125; -100 ], Mid.=[-100; -60] and Surf.=[-60; - 20] m. Vertical lines separate BU (Butterfly) and VM (Virtual Mooring) periods: white lines on the left and black lines on the right.

comprised between 1 and 2 mg l$^{-1}$. Finally, during the VM period, SPMC levels drop (either from acoustics or optics) with
values below 1.5 mg l$^{-1}$.

One can suppose that resuspension processes occurred, caused by the oscillations in the amplitude of the tidal currents, which decreased from spring tide to neap tide from the beginning to the end of the campaign. To examinate this hypothesis, we calculated the current bottom shear stress, and the critical threshold for motion, see e.g. Soulsby (1997). We assumed a logarithmic current profile close to the bottom, with a roughness length $z_0 = k_s/30$, $k_s$ being the Nikuradse roughness length,
and $k_s$ related to the median grain diameter of the bottom superficial sediment, as for instance in (Gentil et al., 2022; Mengual et al., 2019) . Throughout the MELANGE campaign, the shear stress values were on average equal to 0.02 N m$^{-2}$, with a maximum value equal to 0.06 N m$^{-2}$, thus below the critical threshold of 0.08 N m$^{-2}$ for motion, calculated from the Shields parameter. Therefore, turbid waters were probably brought to the study area through advection, from more coastal waters where resuspension was more likely to occur due to favourable hydrosedimentary conditions. This argument is supported by
the fact that the two higher concentration periods correspond to the more coast-ward glider acquisitions. Also, once arrived on the study area, the turbid water was also probably directly advected away, so it had only transited trough the study area during a short period of time, explaining the low concentrations at the end of the campaign.

In summary, the glider successfully captures the temporal and vertical dynamics of SPMC, with consistency between optical and acoustic measurements. Differences observed in SPMC near the seafloor highlight the importance of sensor-specific
sensitivities and emphasize the role of fine particles in nepheloid layers. Additionally the glider demonstrated its capability to provide high-resolution measurements of both barotropic and baroclinic currents with high consistency across platforms. Together, these findings confirm the glider's suitability as a platform for observing hydro-sedimentary processes in coastal zones, providing a solid foundation for process-oriented studies, and allow to compute fluxes.



### 5.3 SPM: linking properties and transport in a tidal shelf environment

In addition to SPMC range, information on the Particle Size Distribution (given in Subsection 5.3.1) is decisive in the study of hydrosedimentary processes that contribute to suspended sediment fluxes observed and presented in Subsection 5.3.2.

#### 5.3.1 Particle size distribution (PSD)

At each vertical bin of the CTD-Rosette casts, the LISST-100X measured $C_v$ [$\mu$l l$^{-1}$]: the volumetric concentration by size classes of particles. Table 6 gives the statistical values of $C_v$, and as well as for the total volumetric concentrations: $C_{v,Tot}$

[$\mu$l l$^{-1}$]. It shows that median values acquired on 14 February are globally twice as those acquired on 18 February, although concentrations show higher maximal values that day. Otherwise, the median diameter $d_{50}$ keeps a typical value round 35 $\mu$m, with only a slight decrease from 14 to 18 February.

Figure 10 displays the corresponding Particle Size Distribution (PSD) in terms of volumetric normalized concentrations $C_{v,norm}$ ($C_v$ divided by $C_{v,Tot}$), along with the turbidity measurements of the CTD-Rosette turbidimeter. The main mode around

35 $\mu$m can be well seen on PSD curves (Figure 10c,f), but a second mode, around 5 $\mu$m also appears. This second mode is only slightly distinguishable on 14 February but stands out clearly on 18 February. It corresponds to a turbid background of very fine grain size sediments superimposed on the main turbidity signal. On 18 February the level of the main turbidity signal decreases (Figure 10d), with a lesser contribution of the main mode at 35 $\mu$m, which highlights the second mode at 5 $\mu$m (Figure 10f).

Figure 10b,e shows that PSDs are globally homogeneous throughout the water column except in the upper part (heights

above 70 m) on 18 February where a population of particles coarser than the main mode gradually appears, its median size growing as the sea surface approaches. This signal likely comes from air bubbles created by wind at the air-water interface (Wang et al., 2011; Vagle et al., 2010), as supported by the maximum wind speed of $\approx$ 20 m s$^{-1}$ recorded during the campaign on that day (Figure 2e).

Finally, at the coarse tail of the distribution, high concentrations appear specifically in the last highest size class ([322;

380] $\mu$m) measured by the LISST-100X. This occurs especially on 18 February. Such extremely high concentrations indicate that particles can be even coarser than the maximum class size measured (Many et al., 2016; Mikkelsen et al., 2005; Traykovski et al., 1999). Such coarse particles detected all along the water column can be those of zooplankton.

We also evidenced changes through the experiment. At the beginning of the experiment, the PSD was fairly unimodal centered between 50 and 100 $\mu$m corresponding to the presence of flocs. Then, we evidenced a change of the PSD toward a

bimodal distribution and the appearance of a fine mode centered around 5 $\mu$m. As we did not evidence resuspension in our study, these fine particles were probably advected from the coastal area with the presence of riverine discharges (Durand et al., 2018). These observations highlight the predominant role of advection in shaping SPM properties, underlining the need to estimate sediment fluxes to better quantify sediment transport dynamics in the coastal zone.



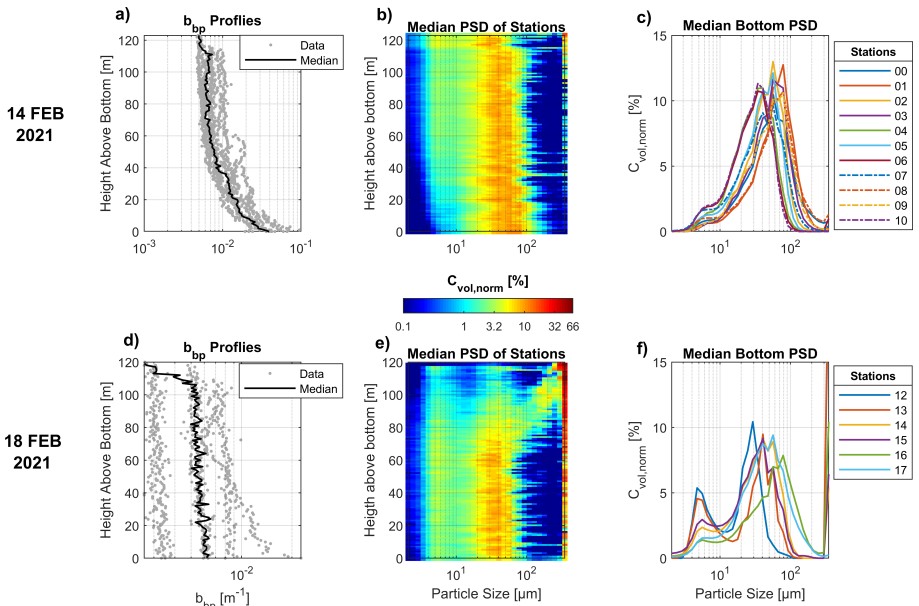

**Figure 10.** Measurements from the CTD-Rosette. **(a)**, **(b)**, and **(c)** Acquisitions on 14 February 2021. **(d)**, **(e)** and **(f)** Acquisitions on 18 February 2021. **(a)** and **(d)** Optical backscattering coefficients ($b_{bp}$) recorded with the FLBBCD triplet.**(b)**, **(c)**, **(e)**, and **(f)** Particle Size Distribution (PSD) recorded with the LISST-100X, with associated normalized volumetric concentrations $C_{v,norm}$. PSDs are obtained in the range 2-380 $\mu$m using 32 logarithmic sampling intervals. Each $C_{vol,norm}$ is computed as the specific volumetric concentration in the considered size range, normalized by the total particulate concentration (over the 32 range classes). **(b)** and **(e)** Median PSD profiles of all stations recorded on 14 and 18 February, respectively. **(c)** and **(f)** Median PSDs close to the bottom (at each station): computed using all LISST-100X measurements recorded at a distance closer than 3 m from the seafloor. The legend indicates the identification numbers of each station. The Y-axis maximum range has been set to 15% for illustration purposes, but the median bottom PSD for stations 12 and 13 (displayed in **(f)**) reach maximum values of 22% and 36% respectively. Locations of all stations are displayed in Figure 1. In **(a)**,**(b)**, **(d)**, **(e)** data were binned from the seafloor level towards the surface so as to express the vertical position of data in height above bottom [m], and thus appropriately show the sediment transport dynamics.

### 5.3.2 Derived SPM fluxes

Estimation of SPM fluxes is still poorly documented at the scale of entire continental shelves. There are almost exclusively single point measurements from bottom tripods, buoys and moorings (Guillén et al., 2006) describing the impact of extreme events at high temporal resolution. Few studies using gliders investigated the sediment transport at the scale of continental shelves (Gentil et al., 2020, 2022; Miles et al., 2015).

Here, thanks to the installation of both a turbidimeter and an AD2CP onboard the SeaExplorer glider, this platform is able
to assess fluxes, in particular the integrated flux, notated $Q_S$, thanks to Equation 3. Figure 11 demonstrates this capacity. The figure shows that the observed instantaneous flux obviously varies with the tide. Amplitudes of fluxes vary from zero to nearly 40 g m$^{-1}$ s$^{-1}$ in the direction of the tidal ellipse. The black and blue curves show that fluxes derived from the acoustics



often underestimate those derived from the optics by around 30%. Then an AD2CP alone onboard a glider could provide an acceptable order of magnitude of such fluxes. However this must require an appropriate calibration of the AD2CP in terms
of SPMC with concomitant SPM filter acquisitions, which is not an obvious task with a moving glider. Here we provided turbidimeters measurements using the same optical instrument model onboard both the glider and our CTD-Rosette deployed in the same area, which allows us to provide an effective calibration (by successive deduction) of both the optical and acoustic sensors onboard the glider. Hence, both a turbidimeter and an AD2CP is certainly mandatory for such flux observation by glider.

To have an idea of the resulting suspended sediment transport in a given direction, residual fluxes are preferable to instantaneous fluxes (the latter being subject to the ebb and flow of the tides). To assess residual fluxes we decided here to apply a basic tidal filter on the observed instantaneous fluxes from the glider. This filter is a 25h window running average applied to both components $U$ and $V$ of the instantaneous fluxes. However, applying such a filter here assumes the fluxes observed by the glider in the whole area are fully consistent in space. This is certainly the case for the currents, as demonstrated by the high
correlation between currents observed by the two moorings GV1 and CIAM, but is not demonstrated in terms of SPMC. Also we can notice on the time series of this Figure that residual fluxes still display some semidiurnal signal, probably because our filter and/or assumption are not perfect. Nonetheless, Figure 12 shows orders of magnitudes round 1 g m$^{-1}$ s$^{-1}$ heading mainly north that are consistent with those modelled for winter conditions in the area of the "Grande Vasière" by Mengual et al. (2019).

Accuracy of suspended sediment fluxes estimates depends on both precision of ocean current measurements and SPMC. If
we define a parameter called 'flux magnitude', notated $M_Q$, with its two components $u$ and $v$ calculated separately as follows:

$$M_{\mathrm{Q}u}(t,z) = \log_{10}(SPMC(t,z)) \cdot u(t,z) \tag{12}$$

Then its typical error measured by the glider, in terms of RMSD, can be estimated as follows (for its two components $u$ and $v$):

$$\mathrm{RMSD}(M_{\mathrm{Q}}) = \mathrm{RMSLD}(SPMC) \cdot \mathrm{RMSD}(u\,\mathrm{or}\,v) \tag{13}$$

This formula is valid because error biases are small compared to their respective RMSD values. Also this formula assumes that $SPMC$ and ($u$ or $v$) are two independent random variables. Then, applying this equation, we take the RMSLD value from Equation (9) and a mean RMSD value of round 3 cm s$^{-1}$ for the currents (from Table 3). Then we obtain the following average error estimation for the 'flux magnitude':

$$\mathrm{RMSD}(M_{\mathrm{Q}}) = 0.11 \times 3.0 = 0.33 \log_{10}(\mathrm{mg} \cdot \mathrm{l}^{-1}) \cdot \mathrm{cm} \cdot s^{-1} \tag{14}$$

Because SPMC error distribution is lognormal and $(u,v)$ error distributions are normal, this unique value of 0.33, on its own, fully describes the error distribution of the 'flux magnitude', and so, by deduction, describes the error distribution of the flux of SPM $Q_{\mathrm{F}}$. However, the range and associated unit of this value are not easy to interpret.



Instead, one may prefer to have a broad idea of the flux errors in terms of relative percent difference (RPD). For that purpose, we calculated the relative percent errors for the fluxes recorded at all comparison cells between CIAM and GLI. To do that, we simply added (because the RPD of the product of independent random variables is the sum of each RPD value) the value of RPD($SPMC$), that is 17%, to all relative percent differences recorded for the currents. Table 7 gives the percentiles of the statistical distribution obtained. Large errors (for percentiles 90 and 95 for instance) correspond to low absolute currents much smaller that 3 cm s$^{-1}$, that cannot be correctly observed with a typical error of 3 cm s$^{-1}$. Nonetheless, the median RPD has a value round 33% which appears acceptable compared to the values ranging from 20% to 600% estimated in Gentil et al. (2020).

The advantage of using gliders is to increase resolution measurements at both spatial and temporal scales and investigate extreme events such as floods and storms. Joined use of glider data and their assimilation in modeling work then strongly increase our interpretation of sediment transport on continental shelves (Estournel et al., 2023). Accuracy of suspended sediment fluxes estimates depends on both precision of ocean current measurements and SPMC. While the accuracy of ocean currents is relatively well constrained, the greatest error comes from SPMC estimation. Both acoustic and optical sensors are widely used but strongly depend on particle size, nature and concentration (Many et al., 2016).

From previous works, the main processes involved in sediment transport on continental shelves are of 2 origins: anthropogenic ones such as bottom trawling (Palanques et al., 2006), installation of offshore wind farms (Vanhellemont and Ruddick, 2014) and natural ones: with floods and storms. While suspended sediment transport estimates outside extreme events remain lower by typically one order of magnitude (Gentil et al., 2020), glider deployments can help estimate sediment transport during these extreme events at the scale of entire continental shelves.

However, analysis of glider data often occurs after the recovery of the instruments at sea a few times later. Using gliders and transmitting data in real time can also have advantages in investigating impacts of human activities on continental shelves. Real-time data could also help managers and decision-makers of a regional area in the evaluation of environmental changes that might affect coastal ecosystem such as marine heat waves, algal blooms, or pollutant dispersion.

## 6 Conclusions

The present study deals with the validation of tidal current, acoustic backscatter, and optical turbidity measurements acquired from the SeaExplorer underwater glider (ALSEAMAR), equipped with a Nortek-AD2CP profiler and Seabird-FLBBCD triplet. Glider-based measurements were acquired during five days in February 2021 in the Bay of Biscay (Atlantic continental shelf) under typical winter conditions. Measurements were successfully validated by comparison with in situ profile data simultaneously acquired nearby from two moored ADCPs, and with optical turbidity and a LISST mounted on CTD-Rosette sampler. The main conclusions of this study are:

1. The AD2CP-SeaExplorer glider system is a suitable platform to monitor the water column-resolute ocean currents over the continental shelf in mesotidal settings. $R^2$ of the glider and mooring profiles are above 0.90, for RMSD in the order of $O(3$ cm s $^{-1})$, that is, of the order of the combined measurement uncertainty on the glider and mooring data.





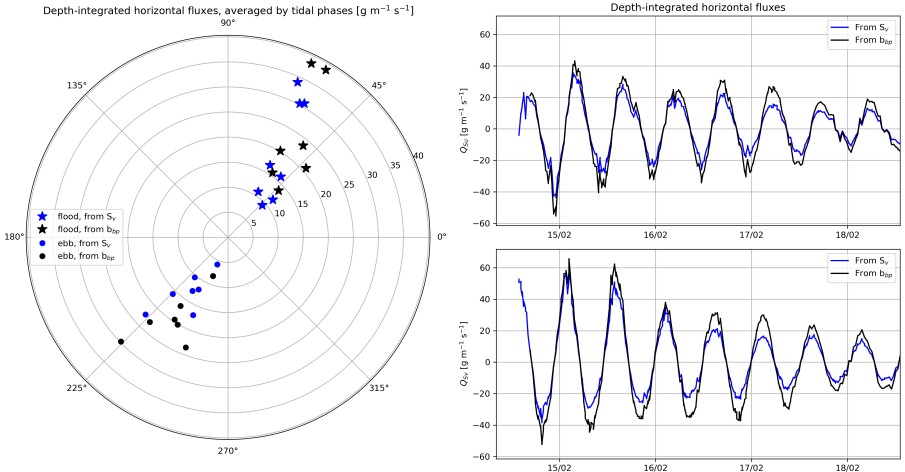

**Figure 11.** Depths-integrated SPMC fluxes derived from the glider currents and concentrations measured either with its AD2CP ($S_v$) or with its FLBBCD sensor ($b_{bp}$). Right: directions and amplitudes of averaged fluxes during full periods of floods (stars) and ebbs (dots); left : time series of the instantaneous fluxes (following the tide mainly). NB: data for $b_{bp}$ starts later due to the missing near bottom data for the first half-day of the campaign.

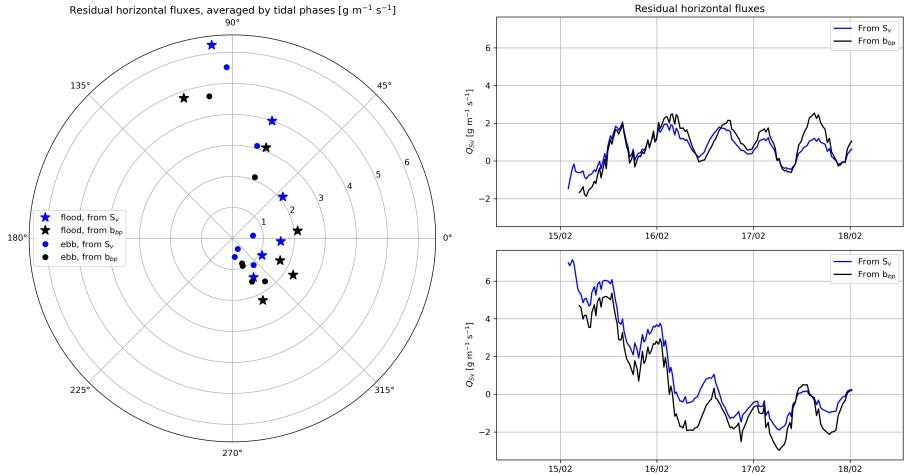

**Figure 12.** Same as Fig.11 except for residual fluxes. NB: due to the 25 hours window of the running average filter (dedicated to remove the semidiurnal tidal signal) the first and last 12.5h data are missing at the beginning and end of sequence (compare to the instantaneous fluxes).

2. In winter conditions, AD2CP-derived acoustic backscatters yield a satisfactory description of the sediment concentration in the nepheloid layer. In combination with optical turbidity, insights on the sediment distribution are also obtainable.



3. In situ calibration of glider-based backscatter sensors with gravimetric measurements will make it possible to accurately estimate suspended particle fluxes at high spatio-temporal scales over the shelf, but this requires a rosette sampler in the vicinity or a similar water-sampling device onboard the glider.


4. During our investigation period, we observed bottom nepheloid layers with SMPC of several mg·l$^{-1}$. Estimated bottom shear stresses do not allow us to conclude to resuspension but rather advection of turbid water waters from coastal area as suggested by estimated SPMC fluxes.

In conclusion, this work provides a promising technological breakthrough for the fine-scale monitoring of ocean water quality

and the outcome of associated suspended matter, for which real-time information on currents and fluxes can also be crucial.

**Abbreviations** The following abbreviations are used in this manuscript:

| | |
|---|---|
| ADCP | Acoustic Doppler Current Profiler |
| $b_{bp}$ | optical backscattering coefficients, derived from the dedicated sensors |
| BU | Butterfly-patterned mobile survey period (realized by the SeaExplorer glider) |
| CU | Combined measurement Uncertainty |
| DAC | Dive Averaged Current |
| IQR | Interquartile Range, such as IQR = Q3 - Q1 |
| LADCP | Lowered ADCP |
| LAT | Lowest Astronomical Tide (chart datum) |
| mab | meters above bottom |
| PSD | Particle Size Distribution |
| Q1 | First quartile |
| Q3 | Third quartile |
| SPM | Suspended Particulate Matter |
| SMPC | Suspended Particulate Matter Concentration |
| STD | Standard Deviation |
| $S_v$ | acoustic backscatter, derived from the glider-AD2CP measurements |
| VM | Virtual Mooring survey period (realized by the SeaExplorer glider) |

*Data availability.* The data are available at https://zenodo.org/records/7866525preview=1&token=eyJhbGciOiJIUzUxMiJ9.eyJpZC
I6IjEyODBiOTRmLWE5M2UtNDRhNS05OTg4LTI3NmVlNDUzODJiYiIsImRhdGEiOnt9LCJyYW5kb20iOiJlYjZkNmU4MTgwMDA
0ZDQ4Y2E0Nzk5ZWE1MjY1ZDBhOSJ9._KggVQzaIqKfLZDtfSdOBjfcI9-AT1yZKtZ2wlc7OVxMc97xOafmjBQ-rxF8FIy9RC0Me__uP
6HiZnV8lT6UWQ , associated with the doi https://doi.org/10.5281/zenodo.7866525 .



*Author contributions.* Conceptualization, F.B. and O.PdF.; methodology, M.G. and O.PdF.; software, O.PdF. and S.H.; validation, S.H.;
formal analysis, S.H. and O.PdF.; investigation, X.DdM.; resources, O.PdF. and F.J.; data curation, S.H. and O.PdF.; writing—original
draft preparation, S.H.; writing—review and editing, F.J., M.G., X.DdM. and F.B.; visualization, S.H.; supervision, F.J. and F.B.; project
administration, F.B.; funding acquisition, F.B. All authors have read and agreed to the published version of the manuscript.

*Competing interests.* The authors declare no conflict of interest. Sponsors had no role in the design of the study; in the collection, analysis
or interpretation of data; in the writing of the manuscript; or in the decision to publish the results.

*Acknowledgements.* Special thanks to our Shom coworkers André Lusven, Vincent Perrier and Tanguy Hermite for the management of
the ADCP acquisitions, ALSEAMAR for providing the glider and its sensor instruments onboard; Laurent Béguery and Vivian Gelas for
the glider deployment; Joelle Salaun, Marine Normant and Sébastien Pinel for the chemistry; Emilien Debonnet for the direction of the
MELANGE campaign onboard RV Thalassa and the GENAVIR staff onboard especially with the quick making of a special fishing net for
the recovering of the glider in rough seas; and to our fellow computer scientists from the Central Web company for the development of the
real-time visualisation and processing software of the SeaExplorer glider data. The authors thank Olivier Peden and the TOI team (LOPS)
for preparing and lending the CIAM bottom mounted ADCP.

*Financial support.* This work was funded by the French DGA (Direction Génerale de l'Armement) and the ANR agency (Agence Nationale
de la Recherche) through the ASTRID-MATURATION MELANGE project (ANR-19-ASMA-0004).





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



| Study and location | glider and profiler model | referencing velocity and validation technique | RMSE cm s$^{-1}$ |
| --- | --- | --- | --- |
| This work Bay of Biscay | SeaExplorer / Nortek AD2CP 1 MHz | BT Two moored RDI 300 kHz | 3 |
| Gentil et al. (2020) NW Mediterranean | Slocum / TRDI DVL Explorer 614 kHz | BT Monte Carlo simulation | 4 (STD) |
| de Fommervault et al. (2019) NW Mediterranean | SeaExplorer / Nortek AD2CP 1 MHz | DAC shipboard ADCP | 2 |
| Ma et al. (2019) South China Sea | Slocum / Nortek AD2CP 1 MHz | DAC Steady modeled velocity | 4 (STD) |
| Todd et al. (2017) West Galapagos | Spray / Nortek AD2CP 1 MHz | DAC, BT, flight model... (*) Crossings of two glider missions | 7-9 |
| Ellis et al. (2015) Off the California Coast | Slocum / TRDI DVL Explorer 614 kHz | DAC One moored RDI 614 kHz | 4-5; 2-3 with an optimal angle of attack |
| Ordonez et al. (2012) Off the Oregon Coast | Slocum / TRDI DVL Explorer 614 kHz | DAC and BT Buoy-mounted RDI 300 kHz | DAC: 6 BT: 4 |
| Thurnherr (2010) North Atlantic Ridge and East Pacific Rise | Vessel-mounted Lowered ADCP | BT One moored ADCP or one CTD-Rosette mounted RDI 300 kHz | 2-3 when sufficient available scatterers |
| Fong and Monismith (2004) Off San Clemente Island, California | Vessel-mounted RDI 600 kHz | BT On a vessel, repeated twice but in opposite directions | 5 |

**Table 4.** Literature review of the uncertainty (RMSE, cm s$^{-1}$ ) obtained in depth-resolved absolute current measured from gliders or vessels. Studies presented here all use the shear method (Visbeck, 2002) to compensate for the glider's motion, except Todd et al. (2017), who uses the inverse method. BT stands for bottom track, DAC for dive-averaged current; (*) see the article by Visbeck (2002) for all the constraints used.



**Table 5.** Statistics of SPMC values derived from optical and acoustic sensors. The optical backscattering coefficients are measured by both the CTD-Rosette (ROS) and the glider (GLI), and the acoustic backscattering coefficient is measured by the glider (GLI). Statistics are given for the whole survey period (All) and for the days of 14 and 18 February 2021. Q1 is the first quartile, Q3 is the third quartile, IQR is the Inter-Quartile Range. The extremum Range is [min-max].

| Date | Platform | Sensor | Parameter | $SPMC$ [mg l$^{-1}$] | | | | |
|------|----------|--------|-----------|--------|-----|-----|-----|--------|
| | | | | Median | Q1 | Q3 | IQR | Range |
| All | ROS | FLBBCD | $b_{bp}$ | 1.4 | 1.1 | 1.9 | 0.7 | 0.6 - 10 |
| All | GLI | FLBBCD | $b_{bp}$ | 1.2 | 1.1 | 1.6 | 0.5 | 0 - 7.8 |
| All | GLI | AD2CP | $S_v$ | 0.5 | 0.3 | 0.7 | 0.4 | 0.2 - 4 |
| 14 Feb | ROS | FLBBCD | $b_{bp}$ | 1.5 | 1.3 | 2.2 | 0.9 | 1 - 10 |
| 14 Feb | GLI | FLBBCD | $b_{bp}$ | 1.6 | 1.5 | 1.9 | 0.4 | 0 - 7.8 |
| 14 Feb | GLI | AD2CP | $S_v$ | 0.8 | 0.6 | 1.1 | 0.5 | 0.3 - 4 |
| 18 Feb | ROS | FLBBCD | $b_{bp}$ | 1.1 | 0.8 | 1.4 | 0.6 | 0.6 - 3.3 |
| 18 Feb | GLI | FLBBCD | $b_{bp}$ | 1.0 | 0.9 | 1.0 | 0.1 | 0 - 1.2 |
| 18 Feb | GLI | AD2CP | $S_v$ | 0.3 | 0.3 | 0.3 | 0.04 | 0.2 - 0.8 |

**Table 6.** Statistical parameters obtained from the LISST measurements onboard the CTD-Rosette stations deployed within the full water column on 14 February and 18 February 2021: with size classes volumetric concentrations $C_v$, total volumetric concentrations $C_{v,Tot}$, and median diameter $d_{50}$.

| Parameter | Unit | Date | Median | Q1 | Q3 | IQR | Range |
|-----------|------|------|--------|----|----|-----|-------|
| | | All | 2.6 | 0.4 | 6.9 | 6.5 | 0-894 |
| $C_v$ | $10^{-2}$ $\mu$l l$^{-1}$ | 14 Feb | 3.1 | 0.4 | 8.8 | 8.4 | 0-323 |
| | | 18 Feb | 1.7 | 0.4 | 4.3 | 3.9 | 0-894 |
| | | All | 142 | 87 | 273 | 187 | 13-284 |
| $C_{v,Tot}$ | $10^{-2}$ $\mu$l l$^{-1}$ | 14 Feb | 169 | 102 | 330 | 228 | 39-2843 |
| | | 18 Feb | 97 | 46 | 192 | 146 | 13-1360 |
| | | All | 36 | 28 | 49 | 21 | 5-275 |
| $d_{50}$ | $\mu$m | 14 Feb | 37 | 30 | 49 | 19 | 14-160 |
| | | 18 Feb | 33 | 26 | 49 | 23 | 5-275 |

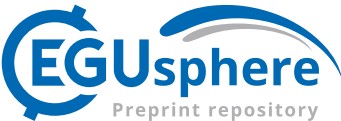

**Table 7.** Percentiles of the statistical distribution of the Relative Percent Difference (RPD) of the instantaneous error flux $Q_F$ based on the difference of the currents recorded between CIAM and GLI.

| Percentile | RPD($Q_F$) in % | |
|---|---|---|
| | U component | V component |
| 10 | 21 | 20 |
| 50 (median) | 34 | 33 |
| 75 | 55 | 45 |
| 90 | 93 | 73 |
| 95 | 154 | 119 |