# Peer review of "Tracing suspended sediment fluxes using a glider: observations in a tidal shelf environment"

_EGUsphere, 2024_

## Referee Comment (RC1)

**Major Comments**

Internal waves (IW) – Authors appear to shy away from presentation and discussion of IW and any spatial variability due to them. I recommend at least presenting the near-bed baroclinic currents at both moorings for comparison, as these are where sediment concentrations are highest. If the near-bed baroclinic currents are very similar then the glider current verification can remain focused on total currents. If not, focus the verification on barotropic currents, but still present RMSD for total and near-bed baroclinic.

Explanation of chosen analyses – There is often quite a few different statistics and measures presented without much explanation as to what the reader should take away from this? I would recommend thinking a bit more about what is the aim of each statistic / analysis you include. For calibration things like R2 are key. For verification of currents things like (total or normalised) RMSD are key. Regression slope could be helpful to show a bias, but regression is not the aim of the verification (I assume) and R2 provides an overly confident estimate to the reader. Comparing statistics such as range or mean speed generally aren't very useful for the verification either.

LISST section - What was the aim of the LISST section? LISST data is very interesting but I wasn't sure what it adds to the paper in its current form. The only takeaway for me was that on 14 Feb when acoustics greatly underestimate optical SPMC near the bed there appeared to be less fines present. I would still include the profiles of $b_{bp}(z)$ if you remove the LISST data.

Schematic – at some points its not exactly clear what is being compared. You could write this clearer in the text but a schematic or photo of the glider with associated measurements might be useful. Given the calibration and current verification is a main aim of the paper referencing another paper is not sufficient.

**Minor comments**

Line 7: 'satisfying' – understandable but probably not quite the right word for a physical sciences manuscript

Line 10: 'from the acoustic' … instrument?

Line 49: More of a limitation for verifying models and not an inherent limitation of our ability to model?

Line 60: Section 2.

Line 60: 'enumerates', describes or details maybe?

Line 90: Remove submesoscale and mesoscale. I would remove these terms from the whole paper and just state distances. I would also give the box dimensions in km.

Line 100: Probably worth extending this sentence about why this BU shape is important.

Line 127: delete notably

Line 152: delete even

Line 158: Change yo to profile or downcast or something, unless this is an accepted glider term.

Line 159: Was stationarity assessed for each 20 min period? Either generally to get a sense if this is appropriate or for the calibration using CIAM mooring data?

Line 175: Is this interpolation step necessary? There is missing data near the surface for the moorings also, why not just stick with observations and not include 'inferred' data?

Line 195: Sentence is a bit confusing, I would say something like 'the  MELANGE area, the instantaneous water level and associated tidal currents are considered…'. I think meso-tidal range is a thing but not mesoscale tidal range.

Line 198: spatially varying bathymetry. It's only varying in time because of the gilder movement.

Line 204: I would expect more than one sentence on this extrapolation, or a highly appropriate reference. In general, I just wouldn't include the extrapolation. Data in Fig5 suggests that current observations do not extend to the sea floor either so that also requires explanation if fluxes are going to be calculated to the floor.

Line 212: Bit strange to throw in specific numbers like 30 um and 1mm for these instruments given it's a continuous function. Could say the 1 MHz scattering response peaks at ~1 mm (10.1016/j.margeo.2010.11.002 Fig 1). But in my experience optical sensors will happily measure coarse sand and the 1 MHz will pick up suspended sediment populations with d50 down to ~20 um (maybe even smaller, I haven't tried). The key assumption you want to make here is that the observed sediment population is not changing much with time, horizontal space, and height above seabed. Maybe this is an opportunity to connect the LISST data?

Line 239: I'm not quite sure how you got the 2.5 cm/s error? Maybe just a standard method? Could compare this to error estimates from similar studies?

Line 254: What does match mean, you assume cells were vertically co-located if the vertical separation was <1m? Or you are interpolating?

Line 255: What does temporally matched mean? Extracting equal / synchronised time points from the datasets or doing some interpolation?

Line 259 and 239: I got a bit confused by the GLI overlap. Does the GLI ADCP record 4 pings per second over 5 seconds (20 pings)? Maybe clearer to say it sampled at 4 Hz continuously and recorded a 5 second average. Then you are computing the expected overlaps based on glider speed correct? Not from raw data? Why less than 3 overlaps at 239 and less than 20% at line 259?

Line 268: delete obviously

Line 280: Were both NTU sensors recently calibrated? Uncalibrated 'identical' instruments can read differently.

Figure 3: Looks like some unresolved dependency on site, i.e. bottom sediment type. Worth mentioning that you expect some of the variability in the relation was probably due to seabed sediment types variation but not included in the regression.

Section 4.9: It is not clear exactly what is being compared here. Are you using the backscatter from just the first ACDP cell to compare to optical? Make clear and state the estimated vertical separation of the optical and acoustic measurements, if any.

Line 304: Doing what?

Line 306: Why are you multiplying the error values by the slope of Eq 9? If the idea to convert acoustic data to SPMC using the 2 regressions in succession then we would expect an increase in uncertainty as error propagates (assuming independent regression models), no?

Line 307: Maybe worth noting that uncertainty will "increase" when converting from log10 units back to normal units. Found example of this here (10.1029/2021JC017538 Fig 12).

Line 326: I don't think you want to be computing the R2 from a linear regression between the 2 moorings (or the glider), if that is what you are doing? You aren't interested in how to translate observations from one thing to another like you were with the calibration, you are interested in the difference (error) between the datasets. This method of computing R2 does not account for magnitude differences between the datasets because it is already accounted for in the slope of the regression. I think if you want a similar metric to your regression R2 you could compute 1 - normalised MSD in a similar way to how you have computed the RMSD (one mooring minus the other). Or maybe you could compute R2 when the regression slope is fixed to 1 and the intercept is fixed to 0. Or just use RMSD and leave out R2 here.

Line 338: Shipwreck seems tenuous. If this is an issue why isn't the RMSD between GV1 and CIAM higher? Seems only an issue with the GV1 – GLI comparison which is difficult to explain when they were so close during the VM period.

Line 343: Systematic bias between GLI and moorings probably not due to spatial variability, rather the shear method as you mention or something else.

Figure 5: black line is free surface – 70 m? Need to somehow mention you have adjusted it for plotting.

Figure 5: This figure isn't really discussed in the text. What is the aim of including it? We can see differences in strength between GLI and moorings, especially at the surface. The white gridded lines make it a bit tough on the eye. Also why the low resolution colormap, was the data too noisy for a good visualisation with a continuous colormap?

Figure 356: Tide was expected to be the main… or show the total + baroclinic currents

Line 358: How should the reader interpret this ratio mean physically? Maybe better to include a bit less here but explain it a bit more.

Line 362: Satisfactory? Also using 'very' is usually avoided for more specific words / numbers.

Line 369: Add figure number again, 'from Figure 7'

Line 375: GLI and CIAM clearly match better in Figure 5. Are you selecting this pair because of this? It is a bit confusing to compare GLI to CIAM when GV1 looks like it was supposed to be the virtual mooring calibration. Is the higher error between GLI and GV1 due to stronger baroclinic currents at GV1 that weren't picked up by the glider very well? If it's because you suspect magnetic interference, or some other error maybe restate it here. But if most of the difference in total currents between GV1 and GLI is found in the baroclinic component there is more to add to the discussion here. If the differences in baroclinic currents are mainly in the BU period, then internal waves are just another source of uncertainty due to spatial separation. If the difference persists through the VM period, then you need to discuss the shortcomings of the glider in observing baroclinic currents.

Line 376: Is +-0.15 for u and +-0.1 for v? Maybe replace first 2 sentences here with GLI u and v std, then CIAM u and v std

Line 378: Stick with the usual RMSD

Line 380: These regions also coincide with the strongest baroclinic currents. If observations from GLI are more uncertain where we want to observe strong baroclinic currents this is a limitation that needs stating.

Line 382: Dispersion of data? Higher uncertainty maybe? Could you add subplot to Fig 8 showing the overlap counts to see if it lines up with errors?

Line 393: Delete obviously

Line 408: Define deep, middle surface here in m as you have in Fig9 caption.

Line 412: You can see the relation deviate from linear in Fig 4 which I think explains why ADCP SPMC is under-estimated at high values. This could be due to a shift to finer particles in theory, but it doesn't look like the LISST supports this. And we would expect a shift towards larger particles as we near the bed, if any change.

Line 414: Need to mention you are moving from muddy to sandy to gravel bottoms over this period. Very important for what you are seeing. The VM phase is on gravel as tides get smaller so maybe not surprising we don't see much then?

Line 424: What is the median grain size? Say it here don't reference it

Line 430: I would remove all the speculation about erosion and advection. This is usually done much better using a bottom lander mooring with turbulence measurements and direct sampling of the seabed. Here you are moving over different sediment types of unknown grain size (at least in this paper) and (I think) extrapolating near bed currents from higher in the water column. Pretty tricky to make accurate comments on erosion and hence the origin of observed sediment. You have observed sediment in suspension and you can calculate some horizontal fluxes, that is the strength of this paper.

Line 435: 'emphasize the role of fine particles in nepheloid layers' – you are bringing back earlier speculation as fact here. This seems to be contradicted in your next section (Fig10 b shows lower volume of fines as you approach the bed).

Line 460: I'm confused as to whether you are saying this is bubbles or zooplankton. I wouldn't speculate, just say it wasn't detected by the NTU sensor and was not considered to be sediment. Could then add that bubbles and biological particles are known to effect LISST.

Fig10b: suggest adding a line that tracks the d50 at each depth going from the surface to the sea floor. Don't include the spikes in the largest bin.

Fig 10d: I wouldn't recommend taking the median of such data that is clearly not grouped. Maybe remove the median and match scatter colors to subplot f?

Fig 10e: Again you have distinct differences in the PSD in subplot f so why take the median here? Maybe show the station with high fines as an example?

Fig 10 caption: Can trim this down. LISST info can be in text or just reference the paper for details. I'd remove the large bins and not discuss values from them here. It is a common issue with the LISST.

Line 478: Because it underestimates SPMC at higher concentrations?

Line 487: I wouldn't expect a 25-hr running mean to remove tides very well. Did you try a lowpass filter like Butterworth?

Line 489: the case for barotropic currents

Line 526: across relevant spatial scales? Lots of gliders to do a whole shelf

Fig 12: Could add filtered currents so reader can see if that's driving the fluxes

Line 539: RMSD is the key metric

---

## Author Comment (AC1)

**Revision of manuscript "Tracing Suspended Sediment Fluxes using a glider: observations in a tidal shelf environment" (EGUSPHERE-2024-4072)**

As a foreword, we would like to extend our sincere thanks to the anonymous referee for his thorough revision of our manuscript, their positive feedback, and constructive comments, which have significantly enhanced the clarity and relevance of our study. We believe the improvements made will deepen the understanding of glider capabilities for monitoring hydro-sedimentary processes in shelf environments. Detailed responses to his comments are provided below (blue text), along with the corresponding changes made to the manuscript.

Please note that, in the revised manuscript, added text appears in magenta colour, and removed text appears in light grey (the same for the Figures and Tables).

**Anonymous Referee #1 ()**

**Major Comments**

1) Internal waves (IW) – Authors appear to shy away from presentation and discussion of IW and any spatial variability due to them. I recommend at least presenting the near-bed baroclinic currents at both moorings for comparison, as these are where sediment concentrations are highest. If the near-bed baroclinic currents are very similar then the glider current verification can remain focused on total currents. If not, focus the verification on barotropic currents, but still present RMSD for total and near-bed baroclinic.

We thank the reviewer for this valuable comment on near-bed baroclinic currents, which helped refine our description of the processes responsible for the discrepancies between glider and mooring currents. In light of this comment Section 5.1 (Validation of glider currents) has been entirely rewritten: the discussion of baroclinic currents has been removed and replaced with a detailed description of bottom boundary layer dynamics to account for the observed vertical patterns of the barotropic components. IW are now explicitly mentioned in the discussion part of Section 5.1.2 (Barotropic current) with two new references added to the article about their generation (Moum et al., 2007; Green et al., 2008). Baroclinic currents can be significant in seasonally stratified environments. However our winter observations show a vertically homogeneous water column: glider's data (see Figure R1 below) measured maximum differences of 0.02°C and 0.01~PSU in the vertical profiles of temperature and salinity respectively). This does not allow for the sustained generation or propagation of baroclinic modes or internal waves. Instead, the near-bed discrepancies between platforms are more readily explained by frictionally driven shear within the bottom boundary layer, generated by the interaction between tidal currents and the seabed. The thickness of this layer, estimated at 9–14 m from the Soulsby, 1983 formulation, decreases in phase with the weakening barotropic current. Moreover, hereafter (see Figure R2 below) are plotted the near-bed residual currents (deviation from the purely vertically barotropic constant) at the two moorings for comparison. It shows the high similarity between them. This is something we checked right from the start of our study.

[Figure]

Fig. R1: profiles of temperature and salinity measured in the whole water column by the glider during all the MELANGE survey period.

[Figure]

Fig. R2: Up) Components of the residual currents (deviation from the purely barotropic constant) measured by the CIAM moored-ADCP. Middle) Components of the residual currents measured by the GV1 moored-ADCP. Down) Components of the difference between the residual currents measured by the two moored-ADCPs. Residual currents are

*defined as u(z,t) - û(t), where û(t) is the strictly barotropic velocity computed from Equation (2). The black straight line delineates the BU and VM survey periods.*

2) **Explanation of chosen analyses** – There is often quite a few different statistics and measures presented without much explanation as to what the reader should take away from this? I would recommend thinking a bit more about what is the aim of each statistic / analysis you include. For calibration things like R2 are key. For verification of currents things like (total or normalised) RMSD are key. Regression slope could be helpful to show a bias, but regression is not the aim of the verification (I assume) and R2 provides an overly confident estimate to the reader. Comparing statistics such as range or mean speed generally aren't very useful for the verification either.

We thank the reviewer for this insightful comment. In particular we well agree that R² alone is not an appropriate metric to assess the agreement between 2 platforms measuring the currents, as it does not account for magnitude differences and may be misleading in this context. Our intention was to evaluate the temporal coherence (or spatial correlation across platforms' data) rather than infer a functional relationship between datasets. To address this, we have fully revised Section 5.1 ("Validation of Glider Currents") to clarify the respective roles of RMSD and R²:

- RMSD is now explicitly used as the primary metric for evaluating intensity agreement between platforms.

- R² is only mentioned to illustrate the similarity in temporal variability (or spatial correlation across platforms' data).

This revision avoids potential misinterpretation and aligns better with the reviewer's recommendation.

3) LISST section - What was the aim of the LISST section? LISST data is very interesting but I wasn't sure what it adds to the paper in its current form. The only takeaway for me was that on 14 Feb when acoustics greatly underestimate optical SPMC near the bed there appeared to be less fines present. I would still include the profiles of bbp(z) if you remove the LISST data.

Thank you for this clarification. We then removed the main LISST data, especially in the corresponding Figure (Figure 11). This Figure has been completely redrawn (as new Figure 12) only keeping some PSD curves near the ocean bottom in order to show the appearance of a second mode of very fine particles at the end of the survey period. Also we kept the profiles of bbp(z) and improved the display in accordance with another comment below. Also the corresponding Section (Section 5.3.2) has been completely rewritten.

4) Schematic – at some points its not exactly clear what is being compared. You could write this clearer in the text but a schematic or photo of the glider with associated measurements might be useful. Given the calibration and current verification, is a main aim of the paper referencing another paper is not sufficient.

We understand you would like us to add an illustration showing all the instruments used (glider, moorings and CTD-rosette) for greater clarity. Actually, such an illustration was included in a previous version of the article but has been removed to save space. Therefore your suggestion confirms that this illustration is lacking. It is now inserted back at the beginning of Section 3 "Materials".

**Minor comments**

Line 7: 'satisfying' – understandable but probably not quite the right word for a physical sciences manuscript

Thank you for the suggestion. We have revised the sentence accordingly to read: (l.6-7)
*"The deployed glider system correctly measures the periodic pattern of the tidal current, with an RMSD of 3 cm s-1, demonstrating the system's ability to accurately capture tidal variability."*

Line 10: 'from the acoustic' … instrument?

Thank you, we replaced "*from the acoustic*" with "f*rom acoustic measurements.*"

Line 49: More of a limitation for verifying models and not an inherent limitation of our ability to model?

Thank you for the clarification. We agree and have revised the sentence as following:
*"This limitation affects our ability to validate sediment transport models, especially under highly dynamic conditions such as storms or floods."*

Line 60: Section 2.1

Thank you for pointing out this writing error. It has now been corrected.

Line 60: 'enumerates', describes or details maybe?

Thank you for the suggestion. We have replaced it with "*describes*".

Line 90: Remove submesoscale and mesoscale. I would remove these terms from the whole paper and just state distances. I would also give the box dimensions in km.

We agree that the terms "mesoscale" and "submesoscale" can be subject to interpretation depending on the definitions used. We have therefore replaced them with explicit distance ranges, as suggested.

Line 100: Probably worth extending this sentence about why this BU shape is important.

Thank you for the suggestion. We extended the sentence to clarify the relevance of the BU shape, highlighting its ability to sample along two orthogonal transects, which improves spatial coverage and captures both along- and cross-track variability (Bosse and Fer, 2019; Rollo et al., 2022; Cauchy et al., 2023).

Line 127: delete notably

It has been removed.

Line 152: delete even

Done.

Line 158: Change yo to profile or downcast or something, unless this is an accepted glider term.

Thank you for the comment. We use "yo" as a standard glider term referring to one full dive–climb cycle. We have clarified this definition at its first occurrence in the manuscript and kept it throughout for consistency. The revised sentence now reads: *"To obtain the ocean contribution as a full water column profile, the LADCP shear method (Visbeck, 2002) was applied to each individual dive–climb cycle (hereafter yo) of the glider."*

Line 159: Was stationarity assessed for each 20 min period? Either generally to get a sense if this is appropriate or for the calibration using CIAM mooring data?

Thank you for the comment. The assumption of current stationarity over the duration of a yo (20 min) was assessed using the CIAM mooring data. As explained in Section 4.6 (L252–255), the standard deviation of horizontal velocities over averaging periods of 30 min was estimated to be 2.1 cm s$^{-1}$, encompassing both measurement uncertainty and ocean variability. This supports the validity of the stationarity assumption over a typical glider yo. A reference to Section 4.6 was also added in the main text where this assumption is introduced.

Line 175: Is this interpolation step necessary? There is missing data near the surface for the moorings also, why not just stick with observations and not include 'inferred' data?

Thank you for pointing out this confusion. The interpolation step near the surface is a standard part of the glider data processing pipeline, intended to reconstruct the top 2–3 m of the water column when needed. However, these extrapolated data were not used in our study. All comparisons with moored instruments were carried out at depths greater than 20 m, and all glider-based results shown in the figures refer to directly observed data. To avoid any confusion, we have removed the following step from the description of the quality control procedure:
*"to linearly extrapolate missing values close to the sea surface."*

Line 195: Sentence is a bit confusing, I would say something like 'the submesoscale' MELANGE area, the instantaneous water level and associated tidal currents are considered...'. I think meso-tidal range is a thing but not mesoscale tidal range.

Thank you for the helpful clarification. We agree that the original phrasing was confusing, and that "meso-tidal range" refers to a tidal regime rather than a spatial scale. To improve clarity, we removed the terms "mesoscale" and "submesoscale" from the main text and replaced them with an explicit distance range. The revised version now reads:
*"In the MELANGE area (10 NM), the instantaneous water level and associated tidal currents are considered spatially homogeneous, as they operate at larger scales than the study area."*

Line 198: spatially varying bathymetry. It's only varying in time because of the gilder movement.

Thank you for the clarification. We agree that the depth variation results from the glider moving across spatially varying bathymetry. We have revised the sentence accordingly to reflect this. The new sentence reads:
*"Bottom depth varied over time along the glider track (−121 m to −105 m) as the glider moved across spatially varying bathymetry."*

Line 204: I would expect more than one sentence on this extrapolation, or a highly appropriate reference. In general, I just wouldn't include the extrapolation. Data in Fig5 suggests that current observations do not extend to the sea floor either so that also requires explanation if fluxes are going to be calculated to the floor.

OK, we added the following precision in this Section 4.3:. because the glider reverses its trajectory when approaching an interface, there are missing data during those phases. Consequently, at a usual distance of about 8 m from the ocean bottom and 16 m from the ocean surface, data are lacking. The missing data (u,v) and SPMC located near the bottom and the surface are completed by extrapolation, repeating the closest value recorded near the interface.

Concerning Fig5, we also precise the following in the figure caption: for the glider data, we can see that bottom depth evolves with the trajectory of the glider. The deepest measurements follow the bottom at a distance of about 8 m. Missing data near the bottom (usually less than 8 m from it) result from reversing of the glider trajectory near the seafloor.

Also, for your information, the Figure R3 below displays the magnitude of the glider currents, with their extrapolated values.

[Figure]

*Fig. R3: profiles of current magnitudes recorded by the glider, and extrapolated to the interfaces. The black dashed line delineates the limit of glider measurements (NB: figure date=real date - 1 day).*

Line 212: Bit strange to throw in specific numbers like 30 um and 1mm for these instruments given it's a continuous function. Could say the 1 MHz scattering response peaks at ~1 mm (10.1016/j.margeo.2010.11.002 Fig 1). But in my experience optical sensors will happily measure coarse sand and the 1 MHz will pick up suspended sediment populations with d50

down to ~20 um (maybe even smaller, I haven't tried). The key assumption you want to make here is that the observed sediment population is not changing much with time, horizontal space, and height above seabed. Maybe this is an opportunity to connect the LISST data?

Thank you for the valuable comment. We agree that backscatter sensitivity depends on both particle size and concentration, and that the previous sentence gave a misleadingly oversimplified interpretation. The corresponding sentence has been removed from the text (beginning of Section 4.4).

As suggested, we now explicitly state in the Discussion (Section 5.3.1) that interpreting acoustic backscatter as a proxy of suspended sediment concentration relies on the assumption that the sediment population remains relatively homogeneous in time and space. This assumption is supported by the LISST measurements, which showed little variability in particle size distribution across the study area.

Line 239: I'm not quite sure how you got the 2.5 cm/s error? Maybe just a standard method? Could compare this to error estimates from similar studies?

Thank you for your comment. We have clarified the method used to derive the $2.5\,\mathrm{cm\,s^{-1}}$ uncertainty in the revised text. This estimate is based on the propagation of independent uncertainties from water pings and bottom-track measurements, each with a nominal ensemble-level precision of $3\,\mathrm{cm\,s^{-1}}$. The resulting horizontal velocity uncertainty is computed using standard error propagation (Eq. 5), assuming the most conservative case with only three overlapping measurements per bin. This yields an upper bound of $2.5\,\mathrm{cm\,s^{-1}}$. These values are consistent with error ranges reported in recent glider-based coastal deployments, such as Ma et al. (2019), who estimate horizontal velocity errors on the order of $4\,\mathrm{cm\,s^{-1}}$.

Line 254: What does match mean, you assume cells were vertically co-located if the vertical separation was <1m? Or you are interpolating?

Thank you for pointing this out. We have clarified the methodology in the revised text.
Vertical "matching" refers to a simple pairing of ADCP and glider cells when their center depths were within 1 m of each other, without interpolation. Similarly, temporal "matching" involved selecting the closest-in-time ADCP profiles to each glider yo, either by averaging over the glider sampling period (CIAM) or selecting the nearest timestamp within a ±30 min window (GV1), again without interpolation. This has now been made explicit in the Methods section.

Line 255: What does temporally matched mean? Extracting equal / synchronised time points from the datasets or doing some interpolation?
See the answer to the previous comment.

Line 259 and 239: I got a bit confused by the GLI overlap. Does the GLI ADCP record 4 pings per second over 5 seconds (20 pings)? Maybe clearer to say it sampled at 4 Hz continuously and recorded a 5 second average. Then you are computing the expected overlaps based on glider speed correct? Not from raw data? Why less than 3 overlaps at 239 and less than 20% at line 259?

Thank you for this detailed comment. We confirm that the AD2CP sampled continuously at 4 Hz and recorded 5-second ensemble averages, as now clarified in the text (L.245-247). The

number of overlapping measurements per depth cell was estimated based on the glider's vertical velocity and sampling interval. The criterion of "less than 3 overlaps" (L.251) refers to a vertical quality threshold applied to AD2CP glider velocity profiles.
In contrast, the "less than 20% overlaps" criterion (L.277) applies to the temporal averaging of mooring data over the glider sampling window and is used to discard cells with insufficient valid data during that period. We have clarified the distinction in the revised manuscript: "*To ensure reliable statistical comparisons between platforms, cells with less than 20\% of overlaps throughout the averaging period, due to failed quality controls, were discarded*."

Line 268: delete obviously
Done.

Line 280: Were both NTU sensors recently calibrated? Uncalibrated 'identical' instruments can read differently.
Thank you very much for this reminder. We then checked the dates of calibration. FLBBCD onboard the CTD was calibrated on 8 January 2021 (1 month before the survey) but FLBBCD onboard the glider was calibrated on 8 July 2015, so 6 years before the survey! In consequence, the end of Sections 4.8 and 4.9 (In situ SPMC calibration…) have been rewritten. Old Section 5.2.1 (Optical and acoustic comparison) was split in two new Sections (5.2.1 Optical sensors comparison and 5.2.2 Acoustic SPM comparison) and rewritten as well. We also checked all the rest of the article in order to verify in particular whether some conclusions could have been overstated in light of this new situation. Fortunately only one sentence, even not decisive, was removed near the beginning of Section 5.3.2.

Figure 3: Looks like some unresolved dependency on site, i.e. bottom sediment type. Worth mentioning that you expect some of the variability in the relation was probably due to seabed sediment types variation but not included in the regression.
Thank you for this precision. We included this comment in the proper Section 4.8 (near the end of this Section).

Section 4.9: It is not clear exactly what is being compared here. Are you using the backscatter from just the first ACDP cell to compare to optical? Make clear and state the estimated vertical separation of the optical and acoustic measurements, if any.
OK, we finally added a sentence towards the end of Section 4.4 and also included the main vertical information in the beginning of Section 4.9.

Line 304: Doing what?
We acknowledge the lack of clarity in the original sentence and have rephrased it as follows: "*To ensure consistency, the surface layer was excluded from the regression, as micro-bubbles and plankton can alter the acoustic-to-optical response and introduce strong variability (Jourdin et al., 2014). The resulting calibration (Sv, bbp), representative of the full MELANGE period, is given by:*"

Line 306: Why are you multiplying the error values by the slope of Eq 9? If the idea to convert acoustic data to SPMC using the 2 regressions in succession then we would expect an increase in uncertainty as error propagates (assuming independent regression models), no?

Thank you for pointing this out. The error values have been revised in accordance with your comment (compute sqrt(x2+y2)), and added in the newly Equation 13 of the article.

Line 307: Maybe worth noting that uncertainty will "increase" when converting from log10 units back to normal units. Found example of this here (10.1029/2021JC017538 Fig 12). We thank the reviewer for this helpful reference. We have added a sentence to acknowledge the increase in relative uncertainty and skewness when converting from logarithmic to linear units. The text now reads: "*Note that converting Equation (11) from logarithmic to linear units increases the relative spread of the uncertainty and skews the resulting distribution, even if no additional error is formally introduced. This effect is discussed in Edge et al. (2022), their Figure 12.*"

Line 326: I don't think you want to be computing the R2 from a linear regression between the 2 moorings (or the glider), if that is what you are doing? You aren't interested in how to translate observations from one thing to another like you were with the calibration, you are interested in the difference (error) between the datasets. This method of computing R2 does not account for magnitude differences between the datasets because it is already accounted for in the slope of the regression. I think if you want a similar metric to your regression R2 you could compute 1 - normalised MSD in a similar way to how you have computed the RMSD (one mooring minus the other). Or maybe you could compute R2 when the regression slope is fixed to 1 and the intercept is fixed to 0. Or just use RMSD and leave out R2 here.
We thank the reviewer for this insightful comment (that also is completing your second major comment). We agree that R² alone is not an appropriate metric to assess the agreement between platforms, as it does not account for magnitude differences and may be misleading in this context. Our intention was to evaluate the temporal coherence rather than infer a functional relationship between datasets. To address this, we have fully revised Section 5.1 ("Validation of Glider Currents") to clarify the respective roles of RMSD and R²:

- RMSD is now explicitly used as the primary metric for evaluating intensity agreement between platforms.

- R² is only mentioned to illustrate the similarity in temporal variability.

This revision avoids potential misinterpretation and aligns better with the reviewer's recommendation.

Line 338: Shipwreck seems tenuous. If this is an issue why isn't the RMSD between GV1 and CIAM higher? Seems only an issue with the GV1 – GLI comparison which is difficult to explain when they were so close during the VM period.
Following this comment we clarified the choice of the metrics involved in Table 4 while removing the parameters a and b, and introducing the simple bias in place of them. Values of biases further support the shipwreck assumption (yet without being able to fully demonstrate it). When biases are high GV1 is involved. Concerning the RMSD, they are a result of both the impact of biases and unbiased variability. The corresponding discussion of all of this has been updated with a text put near the end of this Section 5.1.1 (Total current).

Line 343: Systematic bias between GLI and moorings probably not due to spatial variability, rather the shear method as you mention or something else.

The text has been revised as indicated in the previous comment, removing the reference to spatial variability and attributing the bias to the shear-based reconstruction method.

Figure 5: black line is free surface – 70 m? Need to somehow mention you have adjusted it for plotting.

Thank you for pointing out this detail. Ok we added the precision "put in the middle of the graphic for a better display" in the legends of Figures 7 and 10.

Figure 5: This figure isn't really discussed in the text. What is the aim of including it? We can see differences in strength between GLI and moorings, especially at the surface. The white gridded lines make it a bit tough on the eye. Also why the low resolution colormap, was the data too noisy for a good visualisation with a continuous colormap?

Beyond the usual error metrics, this figure visually confirms the absence of significant artefacts in the glider-derived currents, which follow the same tidal phase and amplitude as the mooring measurements over the full deployment period. Also we can notice that "Sometimes we can see differences in strength between GLI and moorings, especially near the ocean surface", which has been added to the text.

We agree about the poor colormap. We then improved the display in the new Figure 7 of the article. Thank you for the comment.

Figure 356: Tide was expected to be the main… or show the total + baroclinic currents
Line 358: How should the reader interpret this ratio mean physically? Maybe better to include a bit less here but explain it a bit more.

Thank you for this comment. We then rewrote the beginning of this Section 5.1.2 (Barotropic current) and clarified the text in accordance. In particular we removed the ratio values (that are not essential) and explained more instead.

Line 362: Satisfactory? Also using 'very' is usually avoided for more specific words / numbers.

We agree that the wording was imprecise and have revised the sentence for clarity as follow:
"*The RMSD of about $3\,cm\,s^{-1}$ between GLI and CIAM is within the range of the expected Combined Uncertainty, indicating a good agreement between the two independent platforms.*"

Line 369: Add figure number again, 'from Figure 7'
OK

Line 375: GLI and CIAM clearly match better in Figure 5. Are you selecting this pair because of this? It is a bit confusing to compare GLI to CIAM when GV1 looks like it was supposed to be the virtual mooring calibration. Is the higher error between GLI and GV1 due to stronger baroclinic currents at GV1 that weren't picked up by the glider very well? If it's because you suspect magnetic interference, or some other error maybe restate it here. But if most of the difference in total currents between GV1 and GLI is found in the baroclinic component there is more to add to the discussion here. If the differences in baroclinic currents are mainly in the BU period, then internal waves are just another source of uncertainty due to spatial

separation. If the difference persists through the VM period, then you need to discuss the shortcomings of the glider in observing baroclinic currents.

Thank you for this comment since our choice appears not clear enough in the text. Following your comment, and also in accordance with your previous comment on Line 338, the corresponding Section 5.1.3 (Baroclinic current) has been removed and parts completed rewritten and put in new Section 5.1.2 (Barotropic current). In fact the choice is simple, CIAM has been chosen because corresponding biases with GLI are lower. We think that biases are the result of a problem to the compass of GV1 being too close to the Shipwreck Erika, and these biases are an important contribution to the whole RMSD.

Line 376: Is +-0.15 for u and +-0.1 for v? Maybe replace first 2 sentences here with GLI u and v std, then CIAM u and v std

Thank you for pointing out this. As stated just before, the text (of this new Section 5.1.2) has been completely rewritten and is clearer now.

Line 378: Stick with the usual RMSD

Done

Line 380: These regions also coincide with the strongest baroclinic currents. If observations from GLI are more uncertain where we want to observe strong baroclinic currents this is a limitation that needs stating.

Same as previous comments (Lines 338, 375, 376) and your first major comment, the corresponding Section 5.1.3 has been removed, and rewritten parts have been added to the new Section 5.1.2 (Barotropic current). The text is clearer now. In fact there are no baroclinic currents (strictu senso; neither IW). We just observe here frictionally driven shear within the bottom boundary layer.

Line 382: Dispersion of data? Higher uncertainty maybe? Could you add subplot to Fig 8 showing the overlap counts to see if it lines up with errors?

It appears finally that the fewer overlaps (quality control) alone cannot explain the observed differences, as they are more pronounced near the bottom than at the surface. This is what we stated in the complete rewriting of this Section, as mentioned above with Section 5.1.3 (Baroclinic current) being removed and the corresponding text completely rewritten and included in Section 5.1.2 (Barotropic current).

Line 393: Delete obviously

Done

Line 408: Define deep, middle surface here in m as you have in Fig9 caption.

Done

Line 412: You can see the relation deviate from linear in Fig 4 which I think explains why ADCP SPMC is under-estimated at high values. This could be due to a shift to finer particles in theory, but it doesn't look like the LISST supports this. And we would expect a shift towards larger particles as we near the bed, if any change.

Yes we also think that the relation deviates from linear in Fig 4. This is what we state in our new Section 5.2.2 (Acoustic SPMC estimation). The reason is not obvious. In fact, near the

bed primary particles could be larger (due to higher resuspension), but floc sizes could be smaller (due to higher turbulence). Besides that, the LISST readings appear not decisive.

Line 414: Need to mention you are moving from muddy to sandy to gravel bottoms over this period. Very important for what you are seeing. The VM phase is on gravel as tides get smaller so maybe not surprising we don't see much then?
Thanks for the precision. We added this comment to the beginning of Section 5.2.3.

Line 424: What is the median grain size? Say it here don't reference it
Median grain size is 261 μm for muddy sand, 504 μm for sand and 654 μm for gravelly sand (Garlan et al., 2018). Nonetheless, these values had to be finally removed due to the next comment and corresponding answer.

Line 430: I would remove all the speculation about erosion and advection. This is usually done much better using a bottom lander mooring with turbulence measurements and direct sampling of the seabed. Here you are moving over different sediment types of unknown grain size (at least in this paper) and (I think) extrapolating near bed currents from higher in the water column. Pretty tricky to make accurate comments on erosion and hence the origin of observed sediment. You have observed sediment in suspension and you can calculate some horizontal fluxes, that is the strength of this paper.
Thank you for this experience, also consistent with the other reviewer's comment. We finally decided to remove the corresponding paragraph (of this Section 5.2.3).

Line 435: 'emphasize the role of fine particles in nepheloid layers' – you are bringing back earlier speculation as fact here. This seems to be contradicted in your next section (Fig10 b shows lower volume of fines as you approach the bed).
Thank you for reading this. We removed the corresponding sentence that is useless, near the end of Section 5.2.3.

Line 460: I'm confused as to whether you are saying this is bubbles or zooplankton. I wouldn't speculate, just say it wasn't detected by the NTU sensor and was not considered to be sediment. Could then add that bubbles and biological particles are known to affect LISST.
OK, following the new processing of LISST data suggested below (extreme size classes being removed), these assumptions are no longer retained. Furthermore the main LISST data has been removed from the new Figure 12 following the major comment N°3, and this discussion then becomes no longer relevant. The corresponding paragraph has been rewritten. Thank you for the comments.

Fig10b: suggest adding a line that tracks the d50 at each depth going from the surface to the sea floor. Don't include the spikes in the largest bin.
Following the major comment N°3, this Fig10b has been removed. OK for the spikes, also following a comment below, extreme size classes of LISST data has been removed.

Fig 10d: I wouldn't recommend taking the median of such data that is clearly not grouped. Maybe remove the median and match scatter colors to subplot f?
Done

Fig 10e: Again you have distinct differences in the PSD in subplot f so why take the median here? Maybe show the station with high fines as an example?
OK, the PSD for each station is well displayed. For info, this is not the median of a PSD but the median of the measurements selected close to the ocean bottom. It well displays the two modes of fine particle sizes (5 µm) and the larger ones (35 µm).

Fig 10 caption: Can trim this down. LISST info can be in text or just reference the paper for details. I'd remove the large bins and not discuss values from them here. It is a common issue with the LISST.
OK, the new Figure 13 caption has been modified accordingly and shortened: major text has been removed, and part moved in the main article text. Yes, extreme size classes have been removed following Mikkelsen et al., 2005. This processing has now been mentioned in Section 3.4 (CTD-Rosette instrumentation).

Line 478: Because it underestimates SPMC at higher concentrations?
Yes, this precision has been added to the text in parenthesis (near the beginning of Section 5.3.2).

Line 487: I wouldn't expect a 25-hr running mean to remove tides very well. Did you try a lowpass filter like Butterworth?
Effectively a 25h running is not perfect for removing the tidal signal (Shirahata et al., 2016). So we performed a two-pass filter using both a 25h and 13h running average windows. The additional 13h window removes the remaining semidiurnal component (after applying the 25h filter). It appears to work well in our case: see the new Figure 15 displayed in place of the old Figure (Figure 14). Other filters are well known for removing the tidal signal but their running windows are too large for our case: 37h for Doodson's filter, 49h for Munk, 71h for Godin and Demerliac's ones. We would lose a large part of the beginning and end of our time series (which lasts about 96h in total). A low pass filter would remove all signals below the cutoff frequency, including high frequency signals that are not tides, which is not really required here. For instance inertial waves in the area have a period of about 18h.

Line 489: the case for barotropic currents
OK, added in the text.

Line 526: across relevant spatial scales? Lots of gliders to do a whole shelf
OK, very ambitious for sure! This last part of the sentence has been removed.

Fig 12: Could add filtered currents so reader can see if that's driving the fluxes
The following Figures R4-R9 display the filtered currents next to the filtered fluxes. From this display, interpreting the contribution of the currents to the fluxes appears not obvious. It seems there is a complex contribution between currents and turbidity.

Line 539: RMSD is the key metric
Right: R2 has been removed from the conclusion.

[Figure]

Fig. R4: Amplitude of filtered currents (cm/s) with time (Julian day).

[Figure]

Fig. R5: Direction of filtered currents (degrees) with time (Julian day).

[Figure]

Fig. R6: Amplitude of filtered fluxes (g/m/s): Sv (blue) bbp (black).

[Figure]

Fig. R7: Direction of filtered fluxes (degrees): Sv (blue) bbp (black).

[Figure]

*Fig. R8: U component of filtered fluxes (g/m/s): Sv (blue) bbp (black).*

[Figure]

*Fig. R9: V component of filtered fluxes (g/m/s): Sv (blue) bbp (black).*

---

## Author Comment (AC2)

**Revision of manuscript "Tracing Suspended Sediment Fluxes using a glider: observations in a tidal shelf environment" (EGUSPHERE-2024-4072)**

As a foreword, we would like to extend our sincere thanks to Dr. Jay Lee for his thorough revision of our manuscript, their positive feedback, and constructive comments, which have significantly enhanced the clarity and relevance of our study. We believe the improvements made will deepen the understanding of glider capabilities for monitoring hydro-sedimentary processes in shelf environments. Detailed responses to his comments are provided below (blue text), along with the corresponding changes made to the manuscript.

Please note that, in the revised manuscript, added text appears in magenta colour, and removed text appears in light grey (the same for the Figures and Tables).

**Jay Lee, 21 Jun 2025**
**#General Comments**

This manuscript presents a detailed and comprehensive study of hydro-sedimentary dynamics on the French Armorican shelf using a glider equipped with acoustic and optical sensors. The authors validate the measurements of currents and suspended particulate matter concentrations (SPMC) obtained from the instruments mounted on the glider against those from moored ADCPs and CTD-Rosette casts, offering valuable insights into the potential of autonomous platforms for sediment flux estimation.

The study is well-motivated, methodologically sound, and based on a robust dataset. The authors' efforts to calibrate acoustic and optical sensors for SPMC estimation and decompose current signals into barotropic and baroclinic parts are commendable. However, the manuscript suffers from occasional linguistic awkwardness, convoluted sentence structures, and inconsistent terminology, which make the text hard to follow. The scientific content is solid, but the presentation needs refinement. I recommend minor to moderate revision before acceptance.

**#Specific Comments**

Section 4.2 Barotropic velocity computation:

- The description of the barotropic current as a depth-average from −98 m to −52 m is somewhat unclear. Why were these bounds chosen? Are they related to sensor coverage, the tidal boundary layer thickness, or data quality limits?

Thank you for pointing out this. We clarified these boundaries as such in this Section 4.2: "The value 98 is the difference between 110 m (the typical ocean depth recorded) and 12 m (the typical thickness of the bottom boundary layer computed using the Soulsby (1983) formula). The value 52 m corresponds to the maximum range of the CIAM ADCP recorded at the end of the survey, as seen in Figure 7".

Section 5.1.2 and Section 5.1.3:

- The manuscript states that the barotropic component accounts for nearly all of the total current (i.e., ratio ~1). In that case, is the subsequent discussion on the baroclinic component still necessary? It may help to clarify or simplify this section to avoid confusion and to emphasize the dominance of the barotropic signal

We thank the reviewer for this valuable comment on near-bed currents which helped clarify our description. Section 5.1 (Validation of glider currents) has been entirely rewritten: the Section on "baroclinic" currents has been removed and replaced (in the Section on the barotropic currents) with a detailed description of bottom boundary layer dynamics to account for the observed vertical patterns of the barotropic components.

Our winter observations show a vertically homogeneous water column which does not allow for the sustained generation or propagation of truly baroclinic modes. The near-bed discrepancies between platforms are explained by frictionally driven shear within the bottom boundary layer. The thickness of this layer, estimated at 9–14 m from the Soulsby, 1983 formulation, decreases in phase with the weakening barotropic current.

- The lower panel of Figure 8 shows that even in the mid-water column around 100 m depth, where glider data are expected to be most reliable (due to the highest overlap of measurements), the difference in baroclinic currents between glider and mooring still reaches up to ±6 cm/s. This implies that the relative error could be greater than (or equal to) the ground truth (mooring records).

Yes there are possibly large errors remaining sometimes. In the complete rewrite of this new Section 5.1.2 the values of these errors are mentioned explicitly ("enhanced differences are observed near the surface and bottom, reaching up to 0.06 m s-1"). They are located in particular near the bottom where there are less overlaps, but we agree that these overlaps don't fully explain these errors ("However, quality control (overlaps) alone cannot explain the observed differences, as they are more pronounced near the bottom than at the surface."). The origin of these errors is not fully explained.

- It seems the method used to derive the barotropic component is not clearly explained in the manuscript.

OK, this Section 4.2 has been updated accordingly, notably taking into account your previous comment about the values -52 and -98.

Section 5.2.2:

- The author claims that fine sediment was advected from coastal waters but would benefit from further supporting evidence. A reference to satellite imagery, model hindcasts, or literature describing coastal sediment plumes in the region would strengthen the claim.

Thank you for pointing out this possibly overstated claim, also consistent with the other reviewer's comment. We finally decided to remove the corresponding paragraph (of this Section 5.2.3).

Section 5.3.1:

- Line 456 attributes coarse surface signals to bubbles; this is unlikely below 20 m.

Yes, we agree that only gale force winds are able to drive bubbles down to 20 m. We removed this assumption in the rewritten Section 5.1.3 concerning PSD, assuming a possible presence of plankton instead.

- LISST peaks in the highest bin may be artifacts. If claiming biological origin, cite evidence.

Yes, effectively, the LISST data has been reprocessed: extreme size classes have been removed from the PSD following Mikkelsen et al., 2005, so that possible artefacts are being discarded. This processing has now been mentioned in Section 3.4 (CTD-Rosette instrumentation). Please also note that the corresponding Section (5.3.1) has been completely rewritten.

Section 5.3.2:

- The use of a 25-hour low-pass filter to extract the tidal signal may be insufficient, as the right panel of Figure 12 still shows a significant periodic pattern. To more accurately separate tidal and subtidal components, I recommend applying a harmonic analysis (e.g., Foreman, 1977; see https://www.sciencedirect.com/science/article/pii/0309170889900171).

Effectively a 25h running is not perfect for removing the tidal signal (Shirahata et al., 2016). So we performed a two-pass filter using both a 25h and 13h running average windows. The additional 13h window removes the remaining semidiurnal component (after applying the 25h filter). It appears to work well in our case: see the new Figure 15 displayed in place of the old Figure (Figure 14). Other filters are well known for removing the tidal signal but their running windows are too large for our case: 37h for Doodson's filter, 49h for Munk, 71h for Godin and Demerliac's ones. We would lose a large part of the beginning and end of our time series (which lasts about 96h in total). We also tested the harmonic analysis using the following tidal waves: M2, N2, S2, K2, K1, O1, P1, Q1. However the result appears noisy, which is expected in our case, because harmonic analysis expects "beautiful sinusoids" (to say with exact frequency) which can be relevant to time series of water elevation (but not too rich coastal signals), but is less relevant for time series of currents, and even less relevant we guess for time series of turbid fluxes.

**#Interpretation and physical assumptions:**

- Line 188: The thickness of the bottom boundary layer, stated as 9–14 m in the manuscript, might be quite large according to Soulsby (1983).

We double checked our computations and found the same results. The Figure R10 below gives you the full time series obtained. Here are some details of the computation for information. Formulae are based on Soulsby's 1997 book, page 49:

- The water column is homogeneous
- Latitude 47.26°N
- Main tidal wave M2 semidiurnal (used for sigma parameter)
- 8 full tidal cycles during the survey
- vertically integrated currents
- NB: Umin is positive because the tidal current vector rotates anti-clockwise (viewed from above) and the study area is located in the northern hemisphere.

  exemple cycle 2 CIAM: Umax=0.31m/s Umin=0.14m/s sigma=1.4052e-04 rad/s, f=1.4544*10^(-4)*sind(47.26) rad/s

  >> BBL thickness = 0.0038*((0.31*sigma-0.14*f)/(sigma^2 - f^2)) = 13 m

  → GV1: mean 11 m std 2m , CIAM: mean 12 m std 1 m

[Figure]

*Fig. R10: BBL thickness in function of time, from the 2 ADCP moorings measurements.*

- Eq. (3) presents only the u component of sediment flux. Please include the v component or clarify.

Done

- The statement that 1 MHz acoustic sensors are "sensitive to particles of ~1 mm" may be incorrect. Please check this paper (https://www.nortekgroup.com/assets/documents/Monitoring-sediment-concentration-with-acoustic-backscattering-instruments.pdf).

Thank you for this reference. Also, from the other reviewer's comment, it appears that this sentence gave a misleadingly oversimplified interpretation. Finally the corresponding sentence has been removed from the text (near the beginning of Section 4.4).

- The sentence "especially in winter" (Line 300) should be elaborated in details.

Thank you for the question. We clarified by quoting instead "for particles of mineral origin" (at the beginning of Section 4.9).

- Figure 9 shows two high-SPMC events. If the authors interpret these as resulting from advection of suspended particles from coastal sources, it would strengthen the argument to include flow direction or progressive vector plots to illustrate the possible transport pathways.

Thank you again for this question which also is consistent with the other reviewer's comment. It appears that our claim (advection is more likely than resuspension) cannot be fully supported. We finally decided to remove the corresponding paragraph (of this Section 5.2.3).

- Please ensure that Durand et al. (2018) supports the claims described in the manuscript, particularly the statement regarding riverine export of particles (does particle show similar grain size class?)

Indeed this reference does not well explain the origin of these particles (of size round 5 µm). The origin remains unknown. We put at the end of this Section 5.3.1 that "It could eventually be flocculi, although flocculi should have typical sizes between 10 to 20 µm (Lee et al., 2012)".

**#Specifications and formatting inconsistency:**

- Please clarify the operating range of the LISST-100X Type C, which is stated as 2.5-500 µm by the manufacturer, while the manuscript claims 2-380 µm.

OK, the text has been updated in Section 3.4.

- Include schematics or detailed configuration for the mooring setup if it is accessible.

OK, a new Figure (Figure 3) has been included in the article describing the scientific instrumentation used.

- Ensure the consistency of linear regression results throughout the manuscript. Some plots show R² values while others do not. Also, the p-value should be included as the linear regression is used.

OK, R2 has been added in the text (Section 5.1.2 rewritten) corresponding to all panels of Figure 8. Also, p-values have been added for corresponding Figures 4, 5 and 8, and put in the captions. (typically 12~m) (typically 12~m) (typically 12~m) (typically 12~m) (typically 12~m)

- The term "single yo" should be defined for readers unfamiliar with glider-specific terminology

Thank you for the comment. We use "yo" as a standard glider term referring to one full dive–climb cycle. We have clarified this definition at its first occurrence in the manuscript and kept it throughout for consistency. The revised sentence now reads: "*To obtain the ocean contribution as a full water column profile, the LADCP shear method (Visbeck, 2002) was applied to each individual dive–climb cycle (hereafter yo) of the glider.*"

- Why are Tables 4–7 placed after the references instead of being integrated into the main text? It would improve readability and context if these tables were inserted closer to where they are discussed.

Solved (Table 5 was a bit too long and caused a Latex strange formatting!).

- Table and figure formatting is inconsistent: some figure captions refer to left/right incorrectly (e.g., Figure 11), and colorbar units/locations vary. Table 4's caption appears below the table.

Corrected.

- Please standardize equation references (e.g., use consistently "Eq. (9)" rather than a mix of formats in the manuscript).

Corrected.

- Verify the time boundaries (magenta line) in Figure 5 for BU and VM periods. I thought the BU was between the 14th and 18th of February, and VM is after the 18th of February, according to Figure 1b.

Figures have been checked. We confirm that the date that separates BU and VM is 17 February 1320 UTC.

-Several redundant statements are present in the manuscript (e.g., Line 475: 'thanks to

Equation 3'). The authors are encouraged to revise for conciseness and eliminate unnecessary descriptions.

OK, large parts of the text have been rewritten accordingly.

---

## Referee Report (RR1)

[revised manuscript text omitted]

---

## Author Response (AR2)

Dear Dr Matt Rayson,

we are glad that our manuscript is now at the final step of the peer reviewing process. Please find herewith our revised manuscript. Thank you for your last comment about replacing the barotropic and baroclinic terms by more appropriate ones. Indeed it makes the manuscript clearer. We also agree with the last few minor revisions pointed out by the Anonymous Referee #1. The text has been corrected accordingly for all of them. We are specially thankful to the Anonymous Referee #1 for all the revisions he suggested. We agree that they improved consistently the manuscript.

Please note that, in the revised manuscript, added text appears in magenta colour, and removed text appears in light grey (the same for the Figures and Tables)

Yours sincerly,

Dr Sabrina Homrani et al